# Proper Straight Through Estimator: Breaking symmetry promotes convergence to true minimum

## Abstract

In the quantized network, its gradient shows vanishing except for non-differentiable points. The network thus cannot be learned by the standard back-propagation, so that an alternative approach called Straight Through Estimator (STE), which replaces the part of the gradient with a simple differentiable function, is used. While STE is known to work well for learning the quantized network empirically, it has not been established theoretically. A recent study by Yin et al. (2019) has provided theoretical support for STE. However, its justification is still limited to the model in the one-hidden layer network with the binary activation where Gaussian generates the input data, and the true labels are output from the teacher network with the same binary network architecture. In this paper, we discuss the effectiveness of STEs in more general situations without assuming the shape of the input distribution and the labels. By considering the scale symmetry of the network and specific properties of the STEs, we find that STE with clipped Relu is superior to STEs with identity function and vanilla Relu. The clipped Relu STE, which breaks the scale symmetry, may pick up one of the local minima degenerated in scales, while the identity STE and vanilla Relu STE, which keep the scale symmetry, may not pick it up. To confirm this observation, we further present an analysis of a simple misspecified model as an example. We find that all the stationary points are identical with the vanishing points of the cRelu STE gradient, while some of them are not identical with the vanishing points of the identity and Relu STE. Finally we have numerically confirmed the observation for the mixture Gaussian model with various teacher network.

## 1 Introduction

Quantization of the weights and the activations is a promising technique to save the memory and accelerate the inference speed in deep neural networks (DNNs) which have been getting wider and deeper in recent years. There are two main approaches of the quantization for DNNs (Krishnamoorthi (2018)): Post-training quantization (PTQ) and quantization-aware training (QAT). In the PTQ, the pre-trained networks are simply quantized without re-training the model. This approach allows us to achieve the nearly floating point accuracy at 8-bits, while below 8-bits, this results in significant accuracy degradation. Although there have been recent attempts to alleviate the accuracy degradation in PTQ (Banner et al. (2018); Choukroun et al. (2019); Zhao et al. (2019); Kryzhanovskiy et al. (2021)), the QAT, where the quantized weights and activations are trained (e.g., Zhou et al. (2016); Hubara et al. (2017); Rastegari et al. (2016)), usually leads to better accuracy.

The difficulty in QAT is that the weights and the activations are discretized, and intrinsically non-differentiable. If we take the derivatives forcibly, they either vanish or diverge. To avoid this problem, we replace them with the derivatives of some differentiable function in the backward pass only, called the Straight-Through Estimator (STE) (Hinton et al. (2012); Bengio et al. (2013)). Since the replacement leads to bias, it is not always possible to learn the network successfully. However this approach can be applied to the low bits below 8-bits with tolerable accuracy degradation (Zhou et al. (2016); Choi et al. (2018); Esser et al. (2019); Bhalgat et al. (2020)).

Originally, STE was introduced as the replacement with the derivative of the identity function by Hinton et al. (2012). Later, the term "STE" has been extensively used as the replacement with various functions. In the binary network, Bengio et al. (2013) studied the replacement with the derivative of sigmoid function as well as the original STE, and Hubara et al. (2017) used the derivative of the identity function clipped in the region of $|x| \leq 1$. In low-bits network, Zhou et al. (2016) used the quantized gradient after the replacement with the identity function, and Choi et al. (2018), Esser et al. (2019), and Bhalgat et al. (2020) used the the derivative of the clipped Relu (cRelu) for leaning the step size.

## 1.1 MAIN CONTRIBUTION

In this paper, we refer to the proxy of the gradient as the STE gradient [1]. Although STE has been much success in training quantized DNNs empirically, the theoretical justification is very limited to the specific situations. It also remains unclear what differentiable functions should be used for the STE gradient. Our aim is to shed light on a new factor to determine the functions.

Without assuming the shape of the input distribution and the loss function, we first discuss the properties of the three STEs, identity STE, Relu STE, and cRelu STE in one-hidden-layer network with binary activation from the perspective of the intrinsic symmetry. We find that because the identity STE and Relu STE keep the scale symmetry, the STE gradients should be zero for any scale to converge to the true (local) minimum of the loss function, while in the case of the cRelu STE, which breaks the scale symmetry, even if its gradient is not zero at the minimum at some scale and it becomes zero at other scale, the cRelu STE converges to the minimum. Therefore, the back-propagation using the cRelu STE is most likely to converge to the minimum of the loss function among three STEs. This result reveals differences between Relu STE and cRelu STE, which could not be found in the models by Yin et al. (2019) and Long et al. (2021).

Recently, Kunin et al. (2020) discussed the symmetry embedded in non-quantized neural networks, and the effect of its breaking such as the discretization, the weight decay, and the stochasticity during training. We discuss the new effect of breaking symmetry by the proxy of the gradient in training the quantized neural networks.

Next, to confirm this observation, we have studied a similar model of the Gaussian input as discussed in Yin et al. (2019) as an example. We have employed a misspecified model suitable for most practical situations. Unlike the previous study, the true labels are presumed to be generated by the non-quantized network architecture. Consequently, we find that ignoring the scale degeneracy for the weight in front of the activation, all the stationary points are identical with the vanishing points in the cRelu STE gradient, while some of them are not identical with the vanishing points in the identity and the Relu STE gradients. In particular, at the global minimum point, both identity and Relu STE gradients do not vanish. Finally, to confirm it in more general case, we have numerically studied the model of the mixture Gaussian input.

## 1.2 RELATED WORKS

Recently, there have been a few theoretical studies on the justification of the STEs. Shekhovtsov & Yanush (2020) derived STE by the linearization of their proposed estimator in a stochastic binary deep network where the noises are injected before the binary activation to be a smooth network. Cheng et al. (2019) argued that STE can be interpreted as a projected Wasserstein gradient flow under certain conditions. Yin et al. (2019) that inspired our study examined which of the three STE's, identity STE, Relu STE, and cRelu STE, converges to the true minimum in one-hidden-layer network with binarized activation and Gaussian input data. They clarified that if the labels are obtained from a teacher network of the same architecture (Du et al. (2018)), all the three STEs show the non-negative correlation with the population gradient, but only the identity STE gradient does not become zero in a local minimum. This indicates that the back-propagation using either Relu STE or cRelu STE may convergence to the local minimum, while it is impossible to show the convergence to the local minimum using the identity STE. Furthermore, they showed that all the three STEs gradients vanish at the global minimum, indicating that they may achieve the global minimum independent of the choice of the STE, if we start with an appropriate initial value.

---

[1] It is called "coarse" gradient in Yin et al. (2019).

Very recently, Long et al. (2021) have discussed the justification of a class of STEs with certain monotonicity, which generalizes the Relu STE, in one-hidden-layer network for the hinge loss, claiming that the STE gradients vanish at the global minimum.

## 1.3 NOTATIONS

$|\cdot|$ denotes the Euclidean norm of a vector. A capital letter in bold denotes the matrix, e.g., $\boldsymbol{Z}$, and its component is given by the letter with lower indices, e.g., $Z_{ij}$. A small letter in bold denotes the vector, e.g., $\boldsymbol{g}$, and its component is given by the letter with lower indices, e.g., $g_j$. $\boldsymbol{I}_{m \times m}$ is the $m \times m$ identity matrix, and $\boldsymbol{1}_m$ is the $m$-dimensional vector of all ones.

## 2 GENERAL DISCUSSION ON STEs IN ONE-HIDDEN LAYER CNN WITH BINARY ACTIVATION

### 2.1 PRELIMINARIES

The one-hidden layer convolutional neural network with a binary activation function is realized as

$$y(\boldsymbol{Z}, \boldsymbol{v}, \boldsymbol{w}) = \boldsymbol{v}^T \sigma(\boldsymbol{Z}\boldsymbol{w}) = \sum_i v_i \sigma(\sum_j Z_{ij} w_j), \tag{1}$$

where $\sigma(\cdot)$ is the step function,

$$\sigma(x) = \begin{cases} 1 & x > 0, \\ 0 & x \leq 0, \end{cases} \tag{2}$$

$\boldsymbol{Z} \in \mathbb{R}^{m \times n}$ is constructed by deividing the input data, e.g. an image, into $m$ patches with size $n$, and $\boldsymbol{w} \in \mathbb{R}^n$ and $\boldsymbol{v} \in \mathbb{R}^m$ are the trainable weights, i.e. kernels, in the first and second layers, respectively (Du et al. (2018)). We assume that $\boldsymbol{Z}$ is generated by a continuous distribution $P(\boldsymbol{Z})$. Due to the property of the step function, the network is invariant under the scale transformation for $\boldsymbol{w}$, $\boldsymbol{w} \to k\boldsymbol{w} \ (k > 0)$.

Using the loss for each sample, $\ell(y(\boldsymbol{v}, \boldsymbol{w}, \boldsymbol{Z}); y^*)$ with $y^*$ being the labels generated by the true distribution $q(y^*|\boldsymbol{Z})$, we can express the population loss function as

$$L(\boldsymbol{v}, \boldsymbol{w}; y^*) = \mathbb{E}_{\boldsymbol{Z}} \left[ \ell(y(\boldsymbol{v}, \boldsymbol{w}, \boldsymbol{Z}); y^*) \right]. \tag{3}$$

### 2.2 BACK-PROPAGATION AND STE GRADIENTS

The gradients of the loss for each sample w.r.t $\boldsymbol{v}$ and $\boldsymbol{w}$ can be formally expressed as

$$\frac{\partial \ell}{\partial v_i} = \frac{\partial \ell}{\partial y} \frac{\partial y}{\partial v_i} = \frac{\partial \ell}{\partial y} \sigma(\sum_j Z_{ij} w_j), \tag{4}$$

$$\frac{\partial \ell}{\partial w_j} = \sum_i \frac{\partial \ell}{\partial x_i} \frac{\partial x_i}{\partial w_j} = \sum_i \frac{\partial \ell}{\partial y} v_i \delta\left(\sum_{j'} Z_{ij'} w_{j'}\right) Z_{ij}, \tag{5}$$

where $x_i = \sigma(\sum_j Z_{ij} w_j)$ and we used the property that the derivative of the step function is Dirac's delta function, $d\sigma(z)/dz = \delta(z)$. The gradient w.r.t $\boldsymbol{v}$ can be utilized for learning the network, while it is impossible to use the gradient w.r.t $\boldsymbol{w}$ due to the presence of the delta function. Remarkably, both population gradients w.r.t $\boldsymbol{v}$ and $\boldsymbol{w}$ are smooth functions due to its average over the continuous distribution,

$$\mathbb{E}_{\boldsymbol{Z}}\left[\frac{\partial \ell}{\partial v_i}\right] = \int \left(\prod_{i',j'} d\tilde{Z}_{i'j'}\right) P(\{\tilde{Z}_{i'j'}\}) \frac{\partial \ell}{\partial y} \sigma\left(\tilde{Z}_{i1}|\boldsymbol{w}|\right), \tag{6}$$

$$\mathbb{E}_{\boldsymbol{Z}}\left[\frac{\partial \ell}{\partial w_j}\right] = \frac{1}{|\boldsymbol{w}|} \sum_i F_{ij}(\boldsymbol{v}, \boldsymbol{w}, \tilde{Z}_{i1} = 0),$$

$$F_{ij}(\boldsymbol{v}, \boldsymbol{w}, \tilde{Z}_{i1}) \equiv \int \left(\prod_{(i',j') \neq (i,1)} d\tilde{Z}_{i'j'}\right) P(\{\tilde{Z}_{i'j'}\}) \left[\frac{\partial \ell}{\partial y} v_i \sum_{k'} \mathcal{O}^i_{k'j} \tilde{Z}_{ik'}\right], \tag{7}$$

where we used the orthogonal transformation, $\tilde{Z}_{ik'} = \sum_{j'} \mathcal{O}^i_{k'j'} Z_{ij'}$, $Z_{ij'} = \sum_{k'} \mathcal{O}^i_{k'j'} \tilde{Z}_{ik'}$, with $\mathcal{O}^i_{1j'} = w_{j'}/|\boldsymbol{w}|$ and $\sum_{j'} \mathcal{O}^i_{k'j'} \mathcal{O}^i_{l'j'} = \delta_{k',l'}$. We can easily see that the gradient w.r.t. $\boldsymbol{w}$ does not have the component in $\boldsymbol{w}$-direction, $\sum_j w_j \mathbb{E}_{\boldsymbol{Z}} \left[ \frac{\partial \ell}{\partial w_j} \right] = 0$. Its vanishing originally follows from the scale symmetry of the network: Because the loss has the scale symmetry, its gradient in the $\boldsymbol{w}$-direction is obviously zero.

Instead of the population gradient, we use the STE in practice, which replaces the delta function with the derivative of some differentiable function $\mu_{\mathrm{STE}}$, to train the kernel $\boldsymbol{w}$,

$$g_j^{\mathrm{STE}}(\boldsymbol{v}, \boldsymbol{w}, \boldsymbol{Z}) \equiv \sum_i \frac{\partial \ell}{\partial y} v_i \mu'_{\mathrm{STE}} \left( \sum_{j'} Z_{ij'} w_{j'} \right) Z_{ij}. \tag{8}$$

We consider three types of $\mu_{\mathrm{STE}}(x)$: (i) the identity type $\mu_{\mathrm{STE}}(x) = x$, (ii) vanilla ReLU type $\mu_{\mathrm{STE}}(x) = x\sigma(x)$ and (iii) the cReLU type $\mu_{\mathrm{STE}}(x) = x\sigma(x)\sigma(r - x)$ with the upper clipping value $r$. The surrogate back-propagation using the STE is described in Algorithm 1 (Yin et al. (2019)).

---

**Algorithm 1** Surrogate back-propagation using STE for learning one-hidden-layer CNN.

**Input**: initialization $\boldsymbol{v}^0 \in \mathbb{R}^m$, $\boldsymbol{w}^0 \in \mathbb{R}^n$, learning rate $\eta$.
    **for** $t = 0, 1, ,\ldots$ **do**
      $\boldsymbol{v}^{t+1} = \boldsymbol{v}^t - \eta \, \mathbb{E}_{\boldsymbol{Z}} \left[ \frac{\partial \ell}{\partial \boldsymbol{v}} (\boldsymbol{v}^t, \boldsymbol{w}^t; \boldsymbol{Z}) \right]$
      $\boldsymbol{w}^{t+1} = \boldsymbol{w}^t - \eta \mathbb{E}_{\boldsymbol{Z}} \left[ \boldsymbol{g}^{\mathrm{STE}}(\boldsymbol{v}^t, \boldsymbol{w}^t; \boldsymbol{Z}) \right]$
    **end for**

---

The average of the STE gradients over the input data is also expressed as

$$\mathbb{E}_{\boldsymbol{Z}} \left[ g_j^{\mathrm{STE}}(\boldsymbol{v}, \boldsymbol{w}; \boldsymbol{Z}) \right] = \sum_i \int d\tilde{Z}_{i1} \mu'_{\mathrm{STE}} \left( \tilde{Z}_{i1} | \boldsymbol{w} | \right) F_{ij}(\boldsymbol{v}, \boldsymbol{w}, \tilde{Z}_{i1}). \tag{9}$$

Note that in general, the STE gradients have the non-zero component in the $\boldsymbol{w}$-direction due to their partial collapse of the form of the gradient, $\sum_j w_j \mathbb{E}_{\boldsymbol{Z}} \left[ g_j^{\mathrm{STE}}(\boldsymbol{v}, \boldsymbol{w}; \boldsymbol{Z}) \right] \neq 0$.

It is obvious from Eq.(9) that the only difference between the STE gradients with the identity function, Relu, and cRelu is the region of the integration over $\tilde{Z}_{i1}$,

$$\textcolor{red}{\mathbb{E}_{\boldsymbol{Z}} \left[ g_j^{\mathrm{A}}(\boldsymbol{v}, \boldsymbol{w}; \boldsymbol{Z}) \right] = \sum_i \int_{\mathcal{R}_{\mathcal{A}}} d\tilde{Z}_{i1} F_{ij}(\boldsymbol{v}, \boldsymbol{w}, \tilde{Z}_{i1}),} \tag{10}$$

where $A = \mathrm{id}, \mathrm{Relu}, \mathrm{cRelu}$, and the integration region $\mathcal{R}_A$ are given as $\mathcal{R}_{\mathrm{id}} = (-\infty, +\infty)$, $\mathcal{R}_{\mathrm{Relu}} = [0, +\infty)$, and $\mathcal{R}_{\mathrm{cRelu}} = [0, \frac{r}{|\boldsymbol{w}|}]$.

Following the scale invariance of $F_{ij}(\boldsymbol{v}, \boldsymbol{w}, \tilde{Z}_{i1})$ for $\boldsymbol{w}$, $F_{i'j}(\boldsymbol{v}, k\boldsymbol{w}, \tilde{Z}_{i'1}) = F_{i'j}(\boldsymbol{v}, \boldsymbol{w}, \tilde{Z}_{i'1})$ $(k > 0)$, the STE gradients have the following interesting features:

1. The identity and Relu STE gradients have scale invariance for $\boldsymbol{w}$,

$$\mathbb{E}_{\boldsymbol{Z}} \left[ g_j^{\mathrm{id/Relu}}(\boldsymbol{v}, k\boldsymbol{w}; \boldsymbol{Z}) \right] = \mathbb{E}_{\boldsymbol{Z}} \left[ g_j^{\mathrm{id/Relu}}(\boldsymbol{v}, \boldsymbol{w}; \boldsymbol{Z}) \right]. \tag{11}$$

2. The cRelu STE gradient does not have scale invariance for $\boldsymbol{w}$ due to the clipping effect,

$$\mathbb{E}_{\boldsymbol{Z}} \left[ g_j^{\mathrm{cRelu}}(\boldsymbol{v}, k\boldsymbol{w}, r; \boldsymbol{Z}) \right] \neq \mathbb{E}_{\boldsymbol{Z}} \left[ g_j^{\mathrm{cRelu}}(\boldsymbol{v}, \boldsymbol{w}, r; \boldsymbol{Z}) \right], \tag{12}$$

where we have explicitly shown the dependence of the upper clipping value $r$.

3. Instead, the scale transformation of $\boldsymbol{w}$ for the cRelu STE gradient can be compensated by that of $r$:

$$\mathbb{E}_{\boldsymbol{Z}} \left[ g_j^{\mathrm{cRelu}}(\boldsymbol{v}, \boldsymbol{w}, kr; \boldsymbol{Z}) \right] = \mathbb{E}_{\boldsymbol{Z}} \left[ g_j^{\mathrm{cRelu}}(\boldsymbol{v}, \boldsymbol{w}/k, r; \boldsymbol{Z}) \right]. \tag{13}$$

If we take $kr$ with fixed $r$ large enough to cover most of the distribution $P(\boldsymbol{Z})$, the left-hand side is identical with the Relu STE gradient. Here we define the spread of the distribution as $\rho$. When we redefine $\boldsymbol{w}/k$ as $\boldsymbol{w}$ on the right hand, the cRelu STE gradient for $|\boldsymbol{w}| < r/\rho$ is approximately identical with the Relu STE gradient,

$$\mathbb{E}_{\boldsymbol{Z}}\left[g_j^{\text{cRelu}}(\boldsymbol{v}, \boldsymbol{w}, r; \boldsymbol{Z})\right] \simeq \mathbb{E}_{\boldsymbol{Z}}\left[g_j^{\text{Relu}}(\boldsymbol{v}, \boldsymbol{w}; \boldsymbol{Z})\right] \quad \text{for } |\boldsymbol{w}| < r/\rho. \tag{14}$$

This result is intuitively obvious: when $\boldsymbol{w}$ is small enough to keep most of the pre-activation values in the clipped range, the cRelu gradient and the Relu gradient are approximately identical.

The important issue is that there is no guarantee that the weights obtained by the STE back-propagation are the ones at the (local) minimum of the loss function. In other words, at the stationary points, $(\boldsymbol{v}, \boldsymbol{w}) = (\boldsymbol{v}_s, \boldsymbol{w}_s)$, defined as the vanishing points of the population gradient,

$$\mathbb{E}_{\boldsymbol{Z}}\left[\frac{\partial \ell}{\partial v_i}(\boldsymbol{v} = \boldsymbol{v}_s, \boldsymbol{w} = \boldsymbol{w}_s)\right] = 0, \quad \mathbb{E}_{\boldsymbol{Z}}\left[\frac{\partial \ell}{\partial w_j}(\boldsymbol{v} = \boldsymbol{v}_s, \boldsymbol{w} = \boldsymbol{w}_s)\right] = 0, \tag{15}$$

the STE gradient is not zero in general. Here we consider the local minimum point that the STE gradient does not vanish,

$$\mathbb{E}_{\boldsymbol{Z}}\left[g_j^{\text{STE}}(\boldsymbol{v}_s, \boldsymbol{w}_s; \boldsymbol{Z})\right] \neq 0. \tag{16}$$

Note that $(\boldsymbol{v}, \boldsymbol{w}) = (\boldsymbol{v}_s, k\boldsymbol{w}_s)$ for any $k > 0$ is also the local minimum point due to the scale invariance of the loss. In the case of the Relu/identity STEs, their gradient has the same finite value for any scales due to the scale invariance shown in Eq.(11),

$$\mathbb{E}_{\boldsymbol{Z}}\left[g_j^{\text{id/Relu}}(\boldsymbol{v}_s, k\boldsymbol{w}_s; \boldsymbol{Z})\right] = \mathbb{E}_{\boldsymbol{Z}}\left[g_j^{\text{id/Relu}}(\boldsymbol{v}_s, \boldsymbol{w}_s; \boldsymbol{Z})\right] \neq 0. \tag{17}$$

In the case of the cRelu STE, however, the value of its gradient changes as the scale changes due to the breaking of the scale symmetry shown in Eq.(12). If it crosses zero as a function of scale,

$$\mathbb{E}_{\boldsymbol{Z}}\left[g_j^{\text{cRelu}}(\boldsymbol{v}_s, k_0\boldsymbol{w}_s, r; \boldsymbol{Z})\right] = 0 \text{ for some } k_0, \tag{18}$$

the back propagation using cRelu STE can converge at the local minimum as was done by using the population gradient. Therefore, we conjecture that due to breaking scale symmetry, the back-propagation using cRelu STE is the most likely to achieve the (local) minimum of the loss function in the three STEs. This also implies that the cRelu STE is less biased than the other. We confirm the conjecture analytically for the Gaussian input with Relu-type teacher network in Sec.3, and numerically for the mixture Gaussian input with various teacher network in Sec. 4.

Since the above discussion is focused only on symmetry, the conjecture can be generalized to networks with more symmetries in a straightforward way. In that case, if we employ an STE that breaks all of those symmetries instead of cRelu STE, it is more likely to achieve the minimum.

The similar mechanism can be also found in physical phenomena. For instance, this is the case of the ferromagnet, which has rotational symmetry in energy (or Hamiltonian). In the system, if we impose the magnetic field, which breaks the rotational symmetry, the spin states, corresponding to minimum points, become aligned with the same direction.

## 3 ONE-HIDDEN LAYER CNN WITH BINARY ACTIVATION: GAUSSIAN INPUT AND LABELS GENERATED BY NON-QUANTIZED RELU NETWORK

We consider the simple model similar to Yin et al. (2019) given by the Gaussian input with the variance $\hat{\sigma}^2$, $\boldsymbol{Z} \sim P(\boldsymbol{Z}) = \left(\frac{1}{\sqrt{2\pi\hat{\sigma}^2}}\right)^{mn} e^{-\sum_{i,j} \frac{Z_{ij}^2}{2\hat{\sigma}^2}}$. We use the squared loss function, $\ell(y(\boldsymbol{v}, \boldsymbol{w}, \boldsymbol{Z}); y^*(\boldsymbol{v}^*, \boldsymbol{w}^*, \boldsymbol{Z})) = \frac{1}{2}(y - y^*)^2$. Unlike in Yin et al. (2019), the true labels are assumed to be generated by the non-quantized Relu network,

$$y^*(\boldsymbol{v}^*, \boldsymbol{w}^*, \boldsymbol{Z}) = \sum_i v_i^* f_{\text{Relu}}\left(\sum_j Z_{ij} w_j^*\right), \tag{19}$$

with the Relu activation defined as $f_{\text{Relu}}(x) = x\sigma(x)$.

Note that the models are misspecified, i.e., the population loss cannot be zero even at the global minimum. In fact, it is inferred from the accuracy degradation in most practical situations (Hubara et al. (2017)) that the binary networks are considered as misspecified models. On the other hand, if the true labels are generated by the same binarized architecture as was used in Yin et al. (2019), the network becomes a specified model.

As shown below, this misspecified model leads to a more striking difference between three STEs than the specified model. Particularly, the behaviors of three STE gradients at the global minimum are completely different from each other.

### 3.1 POPULATION LOSS, ITS GRADIENT AND STATIONARY POINTS

The population loss $L(\boldsymbol{v}, \boldsymbol{w}; \boldsymbol{v}^*, \boldsymbol{w}^*)$ is derived in Appendix A.1 as

$$
\begin{aligned}
L(\boldsymbol{v}, \boldsymbol{w}; \boldsymbol{v}^*, \boldsymbol{w}^*) &= \mathbb{E}_{\boldsymbol{Z}} \left[ \ell(\boldsymbol{v}, \boldsymbol{w}, \boldsymbol{Z}; \boldsymbol{v}^*, \boldsymbol{w}^*) \right] \\
&= \frac{1}{8} \boldsymbol{v}^T \left( \boldsymbol{I}_{m\times m} + \boldsymbol{1}_m \boldsymbol{1}_m^T \right) \boldsymbol{v} - \frac{\hat{\sigma}|\boldsymbol{w}^*|}{2\sqrt{2\pi}} \boldsymbol{v}^T \left( \cos\varphi \boldsymbol{I}_{m\times m} + \boldsymbol{1}_m \boldsymbol{1}_m^T \right) \boldsymbol{v}^* \\
&\quad + \frac{\hat{\sigma}^2}{4\pi} |\boldsymbol{w}^*|^2 \boldsymbol{v}^{*T} \left( (\pi - 1) \boldsymbol{I}_{m\times m} + \boldsymbol{1}_m \boldsymbol{1}_m^T \right) \boldsymbol{v}^*
\end{aligned}
\tag{20}
$$

where $\varphi$ is the angle between $\boldsymbol{w}^*$ and $\boldsymbol{w}$. Note that the loss is scale invariant for $\boldsymbol{w}$, and thus its derivative satisfies $\boldsymbol{e}_w \cdot \frac{\partial L}{\partial \boldsymbol{w}} = 0$. The stationary points are given by

$$
\frac{\partial L}{\partial \boldsymbol{v}} = 0 \Leftrightarrow \boldsymbol{v} = \frac{2\hat{\sigma}|\boldsymbol{w}^*|}{\sqrt{2\pi}} \left( \cos\varphi \boldsymbol{I}_{m\times m} + \frac{1 - \cos\varphi}{m+1} \boldsymbol{1}_m \boldsymbol{1}_m^T \right) \boldsymbol{v}^*
\tag{21}
$$

$$
\frac{\partial L}{\partial \boldsymbol{w}} = 0 \Leftrightarrow \boldsymbol{v}^T \boldsymbol{v}^* = 0 \text{ or } \varphi = 0 \text{ or } \varphi = \pi
\tag{22}
$$

where we used $\left( \boldsymbol{I}_{m\times m} + \boldsymbol{1}_m \boldsymbol{1}_m^T \right)^{-1} = \boldsymbol{I}_{m\times m} - \frac{1}{m+1} \boldsymbol{1}_m \boldsymbol{1}_m^T$.

There are three stationary points [2] as classified in Appendix A.2:

1. middle saddle point given by $\boldsymbol{v} = \bar{\boldsymbol{v}} \equiv \frac{2\hat{\sigma}|\boldsymbol{w}^*|}{\sqrt{2\pi}} \left( \cos\bar{\varphi} \boldsymbol{I}_{m\times m} + \frac{1 - \cos\bar{\varphi}}{m+1} \boldsymbol{1}_m \boldsymbol{1}_m^T \right) \boldsymbol{v}^*, \varphi = \bar{\varphi} \equiv \arccos \left[ \frac{\left( \boldsymbol{v}^{*T} \boldsymbol{1}_m \right)^2}{-(m+1)(\boldsymbol{v}^*)^2 + \left( \boldsymbol{v}^{*T} \boldsymbol{1}_m \right)^2} \right]$, which satisfies $\bar{\boldsymbol{v}}^T \boldsymbol{v}^* = 0$. This exists if and only if $(m+1)\left( \boldsymbol{v}^* \right)^2 \geq 2 \left( \boldsymbol{v}^{*T} \boldsymbol{1}_m \right)^2$.

2. local minimum if $(m+1)\left( \boldsymbol{v}^* \right)^2 \geq 2 \left( \boldsymbol{v}^{*T} \boldsymbol{1}_m \right)^2$, otherwise saddle point given by $\boldsymbol{v} = \boldsymbol{v}_\pi \equiv \frac{2\hat{\sigma}|\boldsymbol{w}^*|}{\sqrt{2\pi}} \left( -\boldsymbol{I}_{m\times m} + \frac{2}{m+1} \boldsymbol{1}_m \boldsymbol{1}_m^T \right) \boldsymbol{v}^*, \varphi = \pi$.

3. global minimum given by $\boldsymbol{v} = \boldsymbol{v}_0 \equiv \frac{2\hat{\sigma}|\boldsymbol{w}^*|}{\sqrt{2\pi}} \boldsymbol{v}^*, \varphi = 0$. At the global minimum, the population loss does not become zero, $L(\boldsymbol{v} = \boldsymbol{v}_0, \boldsymbol{w} = \boldsymbol{w}^*) = \frac{\hat{\sigma}^2 |\boldsymbol{w}^*|^2 (\pi - 2)}{4\pi} \boldsymbol{v}^{*T} \boldsymbol{v}$.

### 3.2 STE GRADIENT

The STE gradient is given by

$$
\begin{aligned}
g_j^{\text{STE}} &= \sum_i v_i \mu'_{\text{STE}} \left( \sum_{j'} Z_{ij'} w_{j'} \right) Z_{ij} \\
&\quad \times \left( \sum_{i'} v_{i'} \sigma \left( \sum_{j'} Z_{i'j'} w_{j'} \right) - \sum_{i'} v_{i'}^* f_{\text{Relu}} \left( \sum_{j'} Z_{i'j'} w_{j'}^* \right) \right).
\end{aligned}
\tag{23}
$$

---

[2]The definition of the stationary points is not mathematically rigorous in our paper. Even if points are non-differentiable, but the gradient gives zero at the points in the one-side limit, we call them the stationary points.

The converging points by the back-propagation in Algorithm 1 are characterized by vanishing the gradients,

$$\mathbb{E}_{\boldsymbol{Z}}\left[\frac{\partial \ell}{\partial v_i}(\boldsymbol{v}, \boldsymbol{w}; \boldsymbol{Z})\right] = 0, \;\; \mathbb{E}_{\boldsymbol{Z}}\left[g_j^{\mathrm{STE}}(\boldsymbol{v}, \boldsymbol{w}; \boldsymbol{Z})\right] = 0. \tag{24}$$

We discuss the explicit form of the three STE gradients and the associated vanishing points. The detailed derivation of the STE gradients is shown in Appendix A.3-A.5.

### 3.2.1 IDENTITY STE

The identity STE gradient is given by

$$\mathbb{E}_{\boldsymbol{Z}}\left[\boldsymbol{g}^{\mathrm{id}}\right] = \frac{\hat{\sigma}}{\sqrt{2\pi}}\frac{\boldsymbol{w}}{|\boldsymbol{w}|}\left(\boldsymbol{v}^T\boldsymbol{v}\right) - \frac{\hat{\sigma}^2}{2}\boldsymbol{w}^*\left(\boldsymbol{v}^T\boldsymbol{v}^*\right) \tag{25}$$

Combined with Eq.(21), we find that the equation in Eq.(24) has no solution. Therefore, if we use the back-propagation by identity STE, it can not converge to any points.

### 3.2.2 RELU STE

The Relu STE gradient is expressed as

$$\mathbb{E}_{\boldsymbol{Z}}\left[\boldsymbol{g}_{\mathrm{relu}}\right] = \frac{\hat{\sigma}}{2\sqrt{2\pi}}\frac{\boldsymbol{w}}{|\boldsymbol{w}|}\left[\boldsymbol{v}^T\left(\boldsymbol{I}_{m\times m} + \boldsymbol{1}_m\boldsymbol{1}_m^T\right)\boldsymbol{v}\right] - \frac{\hat{\sigma}^2}{2\pi}\frac{|\boldsymbol{w}^*|}{|\boldsymbol{w}|}\boldsymbol{w}\left[\boldsymbol{v}^T\left(-\boldsymbol{I}_{m\times m} + \boldsymbol{1}_m\boldsymbol{1}_m^T\right)\boldsymbol{v}^*\right]$$
$$- \hat{\sigma}^2\left((\pi - \varphi)\boldsymbol{w}^* + \frac{|\boldsymbol{w}^*|\sin\varphi}{|\boldsymbol{w}|}\boldsymbol{w}\right)\frac{\left(\boldsymbol{v}^T\boldsymbol{v}^*\right)}{2\pi}. \tag{26}$$

Combined with Eq.(21), the gradients vanish at the following points:

1. If $\boldsymbol{v}^*$ satisfies $(m+1)\left(\boldsymbol{v}^*\right)^2 \geq 2\left(\boldsymbol{v}^{*T}\boldsymbol{1}_m\right)^2, \varphi = \bar{\varphi} = \arccos\left[\frac{\left(\boldsymbol{v}^{*T}\boldsymbol{1}_m\right)^2}{-(m+1)(\boldsymbol{v}^*)^2+(\boldsymbol{v}^{*T}\boldsymbol{1}_m)^2}\right]$,
   $\boldsymbol{v} = \bar{\boldsymbol{v}} = \frac{2\hat{\sigma}|\boldsymbol{w}^*|}{\sqrt{2\pi}}\left(\cos\varphi\boldsymbol{I}_{m\times m} + \frac{1-\cos\varphi}{m+1}\boldsymbol{1}_m\boldsymbol{1}_m^T\right)\boldsymbol{v}^*$.

2. $\varphi = \pi, \boldsymbol{v} = \bar{\boldsymbol{v}} = \frac{2\hat{\sigma}|\boldsymbol{w}^*|}{\sqrt{2\pi}}\left(-\boldsymbol{I}_{m\times m} + \frac{2}{m+1}\boldsymbol{1}_m\boldsymbol{1}_m^T\right)$.

We find the solutions are identical with two stationary points shown in item 1 and 2 in Sec.3.1, leading to the the saddle point or local minimum, while the global minimum point at $\varphi = 0, \boldsymbol{v} = \boldsymbol{v}_0$ cannot be obtained. Therefore, by using the back-propagation with Relu STE, it always converges to $\varphi = \pi$ if $(m+1)\left(\boldsymbol{v}^*\right)^2 \geq 2\left(\boldsymbol{v}^{*T}\boldsymbol{1}_m\right)^2$, and it does not converge to any points, otherwise.

### 3.2.3 CLIPPED RELU STE

The cRelu STE is given by

$$\mathbb{E}_{\boldsymbol{Z}}\left[\boldsymbol{g}_{\mathrm{crelu}}\right] = \frac{\hat{\sigma}}{2\sqrt{2\pi}}\frac{\boldsymbol{w}}{|\boldsymbol{w}|}\left(1 - e^{-\frac{1}{2}\left(\frac{r}{\hat{\sigma}|\boldsymbol{w}|}\right)^2}\right)\boldsymbol{v}^T\left(\boldsymbol{I}_{m\times m} + \boldsymbol{1}_m\boldsymbol{1}_m^T\right)\boldsymbol{v}$$
$$- \frac{\hat{\sigma}^2}{2\pi}\frac{|\boldsymbol{w}^*|}{|\boldsymbol{w}|}\left(1 - e^{-\frac{1}{2}\left(\frac{r}{\hat{\sigma}|\boldsymbol{w}|}\right)^2}\right)\boldsymbol{w}\left[\boldsymbol{v}^T\left(-\boldsymbol{I}_{m\times m} + \boldsymbol{1}_m\boldsymbol{1}_m^T\right)\boldsymbol{v}^*\right]$$
$$- \hat{\sigma}^2\left\{C(w, \varphi)\frac{\boldsymbol{w}^*}{|\boldsymbol{w}^*|} + S(w, \varphi)\left(\frac{1}{\sin\varphi}\frac{\boldsymbol{w}}{|\boldsymbol{w}|} - \cot\varphi\frac{\boldsymbol{w}^*}{|\boldsymbol{w}^*|}\right)\right\}\left(\boldsymbol{v}^T\boldsymbol{v}^*\right), \tag{27}$$

where $C(|\boldsymbol{w}|, \varphi)$ and $S(|\boldsymbol{w}|, \varphi)$ are given in Eq.(106). At $\varphi = 0, \pi$ they are simplified as $C(|\boldsymbol{w}|, 0) = \frac{|\boldsymbol{w}^*|}{2\pi}\left(\pi - \frac{r}{\hat{\sigma}|\boldsymbol{w}|}e^{-\frac{1}{2}\left(\frac{r}{\hat{\sigma}|\boldsymbol{w}|}\right)^2}\sqrt{2\pi} - \pi\mathrm{erfc}\left(\frac{r}{\sqrt{2}\hat{\sigma}|\boldsymbol{w}|}\right)\right), C(|\boldsymbol{w}|, \pi) = S(|\boldsymbol{w}|, \pi) = S(|\boldsymbol{w}|, 0) = 0$.

Remarkably, the cRelu STE gradient is proportional to the Relu STE gradient in the case of $\boldsymbol{v}^T\boldsymbol{v}^* = 0$ or $\varphi = \pi$:

$$\mathbb{E}_{\boldsymbol{Z}}\left[\boldsymbol{g}_{\mathrm{crelu}}\right]\big|_{\boldsymbol{v}^T\boldsymbol{v}^*=0 \text{ or } \varphi=\pi} = \left(1 - e^{-\frac{1}{2}\left(\frac{r}{\hat{\sigma}|\boldsymbol{w}|}\right)^2}\right)\mathbb{E}_{\boldsymbol{Z}}\left[\boldsymbol{g}_{\mathrm{relu}}\right]\big|_{\boldsymbol{v}^T\boldsymbol{v}^*=0 \text{ or } \varphi=\pi}. \tag{28}$$

As shown in Sec.3.2.2, all the vanishing points in the Relu STE can be found at $\boldsymbol{v}^T \boldsymbol{v}^* = 0$ or $\varphi = \pi$, leading to the saddle point or the local minimum, so that they are also found in the vanishing points of cRelu STE. On the other hand, the behavior of cRelu STE gradient at $\varphi = 0$ is not related to that of Relu STE gradient. The cRelu STE gradient at $\varphi = 0$ is written as

$$
\mathbb{E}_{\boldsymbol{Z}}\left[\boldsymbol{g}_{\mathrm{crelu}}\right] = \frac{\hat{\sigma}}{2\sqrt{2\pi}}\boldsymbol{w}^*\left(1 - e^{-\frac{1}{2}\left(\frac{r}{\sigma|\boldsymbol{w}|}\right)^2}\right)\boldsymbol{v}^T\left\{\frac{1}{|\boldsymbol{w}^*|}\left(\boldsymbol{I}_{m\times m} + \boldsymbol{1}_m\boldsymbol{1}_m^T\right)\boldsymbol{v}\right.
$$
$$
\left. - \frac{2\hat{\sigma}}{\sqrt{2\pi}}\left((-1+\lambda)\,\boldsymbol{I}_{m\times m} + \boldsymbol{1}_m\boldsymbol{1}_m^T\right)\boldsymbol{v}^*\right\}, \tag{29}
$$

where

$$
\lambda = \lambda(|\boldsymbol{w}|) = \frac{\pi - \frac{r}{\hat{\sigma}|\boldsymbol{w}|}e^{-\frac{1}{2}\left(\frac{r}{\hat{\sigma}|\boldsymbol{w}|}\right)^2}\sqrt{2\pi} - \pi\mathrm{erfc}\left(\frac{r}{\sqrt{2}\hat{\sigma}|\boldsymbol{w}|}\right)}{1 - e^{-\frac{1}{2}\left(\frac{r}{\hat{\sigma}|\boldsymbol{w}|}\right)^2}}. \tag{30}
$$

This becomes zero if

$$
\boldsymbol{v} = \frac{2\hat{\sigma}}{\sqrt{2\pi}}|\boldsymbol{w}^*|\left(\boldsymbol{I}_{m\times m} + \boldsymbol{1}_m\boldsymbol{1}_m^T\right)^{-1}\left((-1+\lambda)\,\boldsymbol{I}_{m\times m} + \boldsymbol{1}_m\boldsymbol{1}_m^T\right)\boldsymbol{v}^*
$$
$$
= \frac{2\hat{\sigma}}{\sqrt{2\pi}}|\boldsymbol{w}^*|\left((-1+\lambda)\,\boldsymbol{I}_{m\times m} + \frac{2-\lambda}{m+1}\boldsymbol{1}_m\boldsymbol{1}_m^T\right)\boldsymbol{v}^*. \tag{31}
$$

To be consistent with Eq.(21) at $\varphi = 0$, only $\lambda = 2$ is allowed. The solution provides the global minimum shown in item 3 in Sec.3.1.

In fact, $\lambda$ monotonically increases from 0 to $\pi$ as $\frac{r}{\hat{\sigma}|\boldsymbol{w}|}$ increases, and thus the solution at $\lambda = 2$, corresponding to the global minimum point, can be obtained for any clipping value $r$ and the variance $\hat{\sigma}^2$ by changing the scale $|\boldsymbol{w}|$. Consequently, we get all the stationary points are found in cRelu STE:

1. If $\boldsymbol{v}^*$ satisfies $(m+1)\left(\boldsymbol{v}^*\right)^2 \geq 2\left(\boldsymbol{v}^{*T}\boldsymbol{1}_m\right)^2$, $\varphi = \bar{\varphi} = \arccos\left[\frac{\left(\boldsymbol{v}^{*T}\boldsymbol{1}_m\right)^2}{-(m+1)(\boldsymbol{v}^*)^2+(\boldsymbol{v}^{*T}\boldsymbol{1}_m)^2}\right]$,
   $\boldsymbol{v} = \bar{\boldsymbol{v}} = \frac{2\hat{\sigma}|\boldsymbol{w}^*|}{\sqrt{2\pi}}\left(\cos\varphi\boldsymbol{I}_{m\times m} + \frac{1-\cos\varphi}{m+1}\boldsymbol{1}_m\boldsymbol{1}_m^T\right)\boldsymbol{v}^*$.

2. $\varphi = \pi, \boldsymbol{v} = \bar{\boldsymbol{v}} = \frac{2\hat{\sigma}|\boldsymbol{w}^*|}{\sqrt{2\pi}}\left(-\boldsymbol{I}_{m\times m} + \frac{2}{m+1}\boldsymbol{1}_m\boldsymbol{1}_m^T\right)$.

3. $\varphi = 0, |\boldsymbol{w}| = w_0 \equiv \frac{r}{\hat{\sigma}}c_0, \boldsymbol{v} = \boldsymbol{v}_0 \equiv \frac{2\hat{\sigma}|\boldsymbol{w}^*|}{\sqrt{2\pi}}\boldsymbol{v}^*$, where $w_0$ (or $c_0$) is given by $\lambda(w_0) = 2$.

Note that while the global minimum point is degenerated in scales for $\boldsymbol{w}$, i.e. if $(\boldsymbol{w}, \boldsymbol{v}) = (\boldsymbol{w}_0, \boldsymbol{v}_0)$ gives the global minimum, $(\boldsymbol{w}, \boldsymbol{v}) = (k\boldsymbol{w}_0, \boldsymbol{v}_0)$ for any $k > 0$ also gives the global minimum, the cRelu STE picks up the point at the scale determined by $\lambda = 2$.

## 4    EXPERIMENTS

To confirm our conjecture in more general case, we have numerically studied the Gaussian mixture input with various mean values. As teacher networks, we have tested tanh-type and sin-type as well as Relu-type. For all the examined setups, cRelu STE behaves like the population gradient, while id/Relu STEs show qualitatively different behaviors as described below. To calculate the population loss and STE gradients, we have generated the mixture Gaussian samples and have taken their average. The population gradient has been obtained by calculating the finite difference of the population loss. We have demonstrated the back propagation with learning rate $\eta = 0.01$ given in Algorithm 1.

Shown in Fig. 1 are the results of ten mixture Gaussian input with random mean values for each components of $\boldsymbol{Z}$. We employed $m = 20, n = 25$, and the tanh-type teacher network. We find that the population gradient and cRelu STE show similar results, while id/Relu STEs are completely different. This indicates cRelu STE is less biased than id/Relu STEs.

In the case of id/Relu STEs, at early steps up to around 500 step, $|\boldsymbol{w}|$ decreases around the magnitude of the update quantity, so that it begins to oscillate. Then it escapes the oscillation, and the loss function shows the convergence to a point different from the local minimum achieved by population gradient, while $|\boldsymbol{w}|$ becomes larger and larger due to their scale invariance. Interestingly, the values of the loss function are small compared to the one obtained by population gradient, which implies that id/Relu STEs avoid being trapped in the local solution due to their large bias. However, note that even at that point, the magnitude of $w$ continues to grow and eventually become numerically unstable.

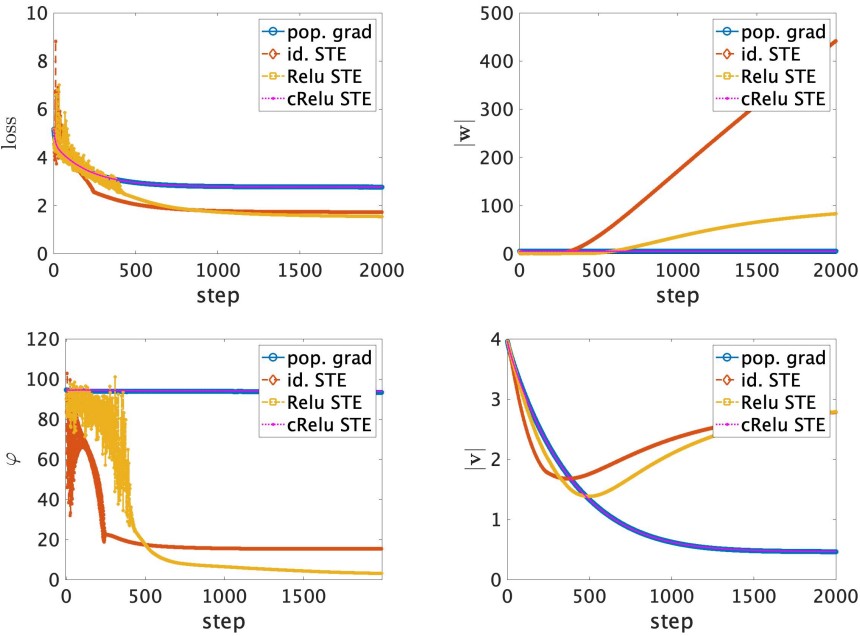

Figure 1: Numerical results of the back-propagation by population gradient and three STEs. We generate 10000 samples which follow ten mixture Gaussian input with random mean values. We employ the tanh-type teacher network.

## 5 SUMMARY

We have found that breaking symmetry embedded in the network by STEs enhances the possibility of convergence to the true (local) minimum of the loss function. We have demonstrated that if an STE breaks the scale symmetry embedded in the one-hidden-layer network with a binary activation, it is more likely to achieve the local minimum than the one which keeps the symmetry. The discussion can be generalized to the network with more symmetries in a straightforward way. The more symmetries embedded in the network an STE breaks, the more likely it is to converge. To confirm the mechanism, we have studied three STEs, identity STE, Relu STE, and cRelu STE, in a simple misspecified model with Gaussian input. We have found that the back-propagation by the cRelu STE, which breaks the scale symmetry, can converge to the global minimum, while identy/Relu STE cannot. Finally we have numerically confirmed the mechanism for the mixture Gaussian model with various teacher network.

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

## A   DERIVATION OF SEC.3

In the Appendix, we provide the results of the Gaussian with variance $1, \mathcal{N}(0, 1)$. For some function $f(\boldsymbol{Z})$, the expectation value of the Gaussian with the generic variance $\hat{\sigma}, \mathcal{N}(0, \hat{\sigma}^2)$, shown in Sec.3 is written as

$$
\begin{aligned}
E_{\boldsymbol{Z} \sim \mathcal{N}(0, \sigma^2)}[f(\boldsymbol{Z})] &= \sqrt{\frac{1}{2\pi\hat{\sigma}^2}} \int \prod_{i,j} dZ_{ij} f(\boldsymbol{Z}) e^{-\frac{\sum_{i,j} Z_{ij}^2}{2\hat{\sigma}^2}} \\
&= \sqrt{\frac{1}{2\pi}} \int \prod_{i,j} dZ'_{ij} f(\hat{\sigma}\boldsymbol{Z}') e^{-\frac{\sum_{i,j} Z_{ij}'^2}{2}} \\
&= E_{\boldsymbol{Z} \sim \mathcal{N}(0, 1)}[f(\hat{\sigma}\boldsymbol{Z})]
\end{aligned}
\tag{32}
$$

where we have changed the integral variables from $\boldsymbol{Z}$ to $\boldsymbol{Z}' = \boldsymbol{Z}/\sigma$. Therefore we get the results with the variance $\hat{\sigma}$ by only changing from $f(\boldsymbol{Z})$ to $f(\hat{\sigma}\boldsymbol{Z})$.

### A.1   DERIVATION OF LOSS FUNCTION

The expectaion of the loss is divided into the following three terms:

$$
\begin{aligned}
L(\boldsymbol{v}, \boldsymbol{w}; \boldsymbol{v}^*, \boldsymbol{w}^*) =& \frac{1}{2} \mathbb{E}_{\boldsymbol{Z}} \left[ \boldsymbol{v}^T \sigma(\boldsymbol{Z}\boldsymbol{w}) \boldsymbol{v}^T \sigma(\boldsymbol{Z}\boldsymbol{w}) \right] - \mathbb{E}_{\boldsymbol{Z}} \left[ \boldsymbol{v}^T \sigma(\boldsymbol{Z}\boldsymbol{w}) \boldsymbol{v}^{*T} f_{\text{Relu}}(\boldsymbol{Z}\boldsymbol{w}^*) \right] \\
&+ \frac{1}{2} \mathbb{E}_{\boldsymbol{Z}} \left[ \boldsymbol{v}^{*T} f_{\text{Relu}}(\boldsymbol{Z}\boldsymbol{w}^*) \boldsymbol{v}^{*T} f_{\text{Relu}}(\boldsymbol{Z}\boldsymbol{w}^*) \right]
\end{aligned}
\tag{33}
$$

The calculation of the first term and the third term are given in "Proof of Lemma 1" in Yin et al. (2019) and "Proof of Section 3" in Ref. Du et al. (2018), respectively:

$$
\begin{aligned}
&\frac{1}{2} \mathbb{E}_{\boldsymbol{Z}} \left[ \boldsymbol{v}^T \sigma(\boldsymbol{Z}\boldsymbol{w}) \boldsymbol{v}^T \sigma(\boldsymbol{Z}\boldsymbol{w}) \right] + \frac{1}{2} \mathbb{E}_{\boldsymbol{Z}} \left[ \boldsymbol{v}^{*T} f_{\text{Relu}}(\boldsymbol{Z}\boldsymbol{w}^*) \boldsymbol{v}^{*T} f_{\text{Relu}}(\boldsymbol{Z}\boldsymbol{w}^*) \right] \\
&= \frac{1}{8} \boldsymbol{v}^T \left( \boldsymbol{I}_{m \times m} + \boldsymbol{1}_m \boldsymbol{1}_m^T \right) \boldsymbol{v} + \frac{1}{4\pi} (\boldsymbol{w}^*)^2 \boldsymbol{v}^{*T} \left( (\pi - 1) \boldsymbol{I}_{m \times m} + \boldsymbol{1}_m \boldsymbol{1}_m^T \right) \boldsymbol{v}^*
\end{aligned}
\tag{34}
$$

To discuss the second term, we evaluate the following quantity:

$$
\begin{aligned}
F_{ij} &= \mathbb{E}_{\boldsymbol{Z}} \left[ \sigma(\boldsymbol{Z}\boldsymbol{w}) f_{\text{Relu}}(\boldsymbol{Z}\boldsymbol{w}^*) \right]_{ij} \\
&= \left( \frac{1}{\sqrt{2\pi}} \right)^{mn} \int \left( \prod_{k,l} dZ_{kl} \right) e^{-\frac{1}{2} \sum_{k,l} Z_{kl}^2} \sigma \left( \sum_{m'} Z_{im'} w_{m'} \right) \left( \sum_{m'} Z_{jm'} w_{m'}^* \right) \sigma \left( \sum_{m'} Z_{jm'} w_{m'}^* \right)
\end{aligned}
\tag{35}
$$

(i) $i \neq j$ case:

$$
\begin{aligned}
F_{ij} &= \left( \frac{1}{\sqrt{2\pi}} \right)^{2n} \int \left( \prod_l dZ_{il} dZ_{jl} \right) e^{-\frac{1}{2} \sum_l (Z_{il}^2 + Z_{jl}^2)} \sigma \left( \sum_{m'} Z_{im'} w_{m'} \right) \left( \sum_{m'} Z_{jm'} w_{m'}^* \right) \sigma \left( \sum_{m'} Z_{jm'} w_{m'}^* \right) \\
&= \left( \frac{1}{\sqrt{2\pi}} \right)^{2n} \int \left( \prod_l d\tilde{Z}_{il} d\tilde{Z}_{jl} \right) e^{-\frac{1}{2} \sum_l (\tilde{Z}_{il}^2 + \tilde{Z}_{jl}^2)} \sigma \left( \tilde{Z}_{i1} |\boldsymbol{w}| \right) \left( \tilde{Z}_{j1} |\boldsymbol{w}^*| \right) \sigma \left( \tilde{Z}_{j1} |\boldsymbol{w}^*| \right) \\
&= \left( \frac{1}{\sqrt{2\pi}} \right)^2 \int d\tilde{Z}_{i1} d\tilde{Z}_{j1} e^{-\frac{1}{2} (\tilde{Z}_{i1}^2 + \tilde{Z}_{j1}^2)} \sigma \left( \tilde{Z}_{i1} \right) \left( \tilde{Z}_{j1} |\boldsymbol{w}^*| \right) \sigma \left( \tilde{Z}_{j1} \right) \\
&= \frac{1}{2\sqrt{2\pi}} |\boldsymbol{w}^*|,
\end{aligned}
\tag{36}
$$

where we used the orthogonal transformation in the second line:

$$
\begin{aligned}
\tilde{Z}_{il} &= \sum_{m'} \mathcal{O}_{lm'}^i Z_{im'}, \\
\tilde{Z}_{jl} &= \sum_{m'} \mathcal{O}_{lm'}^j Z_{jm'}
\end{aligned}
\tag{37}
$$

with $\mathcal{O}^i_{1m'} = w_{m'}/|\boldsymbol{w}|$ and $\mathcal{O}^j_{1m'} = w^*_{m'}/|\boldsymbol{w}^*|$.

(ii) $i = j$ case:

$$F_{ii} = \left(\frac{1}{\sqrt{2\pi}}\right)^n \int \prod_l dZ_{il} e^{-\frac{1}{2}\sum_l Z_{il}^2} \sigma\left(\sum_m Z_{im} w_m\right)\left(\sum_m Z_{im} w^*_m\right) \sigma\left(\sum_m Z_{im} w^*_m\right). \quad (38)$$

Without loss of generality, we can choose

$$\begin{aligned}
\boldsymbol{w}^* &= (w^*, \boldsymbol{0}_{n-1}), \\
\boldsymbol{w} &= (w_1, w_2, \boldsymbol{0}_{n-2})
\end{aligned} \quad (39)$$

with $w^* > 0$, so that this can be written as

$$\begin{aligned}
F_{ii} &= \left(\frac{1}{\sqrt{2\pi}}\right)^2 \int dZ_{i1} dZ_{i2} e^{-\frac{1}{2}\left(Z_{i1}^2 + Z_{i2}^2\right)} \sigma\left(\sum_{m=1}^2 Z_{im} w_m\right) Z_{i1} w^* \sigma(Z_{i1}) \\
&= \left(\frac{1}{\sqrt{2\pi}}\right)^2 \int dZ d\theta Z e^{-\frac{1}{2}Z^2} \sigma\left(\cos(\varphi - \theta)\right) Z \cos\theta w^* \sigma(\cos\theta) \\
&= \frac{1}{2\pi} \int_0^\infty dZ \int_{\varphi - \pi/2}^{\pi/2} d\theta Z^2 e^{-\frac{1}{2}Z^2} \cos\theta w^* \\
&= \frac{1}{2\sqrt{2\pi}} w^*(1 + \cos\varphi)
\end{aligned} \quad (40)$$

where $\varphi$ is the angle between $\boldsymbol{w}^*$ and $\boldsymbol{w}$. Eqs.(36) and (40) are combined as

$$F_{ij} = \frac{1}{2\sqrt{2\pi}} |\boldsymbol{w}^*|(1 + \cos\varphi)\delta_{ij} + \frac{1}{2\sqrt{2\pi}} |\boldsymbol{w}^*|(1 - \delta_{ij}), \quad (41)$$

so that the second term in Eq. (33) is expressed as

$$\mathbb{E}_{\boldsymbol{Z}}\left[\boldsymbol{v}^T \sigma(\boldsymbol{Z}\boldsymbol{w})\boldsymbol{v}^{*T} f_{\text{Relu}}(\boldsymbol{Z}\boldsymbol{w}^*)\right] = \frac{|\boldsymbol{w}^*|}{2\sqrt{2\pi}} \boldsymbol{v}^T \left(\cos\varphi \boldsymbol{I}_{m\times m} + \boldsymbol{1}_m \boldsymbol{1}_m^T\right) \boldsymbol{v}^*. \quad (42)$$

Consequently, the loss function is summarized as

$$\begin{aligned}
L(\boldsymbol{v}, \boldsymbol{w}; \boldsymbol{v}^*, \boldsymbol{w}^*) &= \frac{1}{8}\boldsymbol{v}^T \left(\boldsymbol{I}_{m\times m} + \boldsymbol{1}_m \boldsymbol{1}_m^T\right) \boldsymbol{v} - \frac{|\boldsymbol{w}^*|}{2\sqrt{2\pi}} \boldsymbol{v}^T \left(\cos\varphi \boldsymbol{I}_{m\times m} + \boldsymbol{1}_m \boldsymbol{1}_m^T\right) \boldsymbol{v}^* \\
&\quad + \frac{1}{4\pi}|\boldsymbol{w}^*|^2 \boldsymbol{v}^{*T} \left((\pi - 1)\boldsymbol{I}_{m\times m} + \boldsymbol{1}_m \boldsymbol{1}_m^T\right) \boldsymbol{v}^*.
\end{aligned} \quad (43)$$

The population gradient w.r.t $\boldsymbol{v}$ and $\boldsymbol{w}$ is thus given by

$$\frac{\partial L}{\partial \boldsymbol{v}} = \frac{1}{4}\left(\boldsymbol{I}_{m\times m} + \boldsymbol{1}_m \boldsymbol{1}_m^T\right) \boldsymbol{v} - \frac{\hat{\sigma}|\boldsymbol{w}^*|}{2\sqrt{2\pi}}\left(\cos\varphi \boldsymbol{I}_{m\times m} + \boldsymbol{1}_m \boldsymbol{1}_m^T\right) \boldsymbol{v}^*, \quad (44)$$

$$\frac{\partial L}{\partial \boldsymbol{w}} = \frac{1}{|\boldsymbol{w}|}\frac{\partial L}{\partial \varphi} \boldsymbol{e}_\varphi = \frac{\hat{\sigma}|\boldsymbol{w}^*|}{2\sqrt{2\pi}|\boldsymbol{w}|} \sin\varphi \left(\boldsymbol{v}^T \boldsymbol{v}^*\right) \boldsymbol{e}_\varphi, \quad (45)$$

where without loss of generality, we can take $\boldsymbol{e}_w = (\cos\varphi, \sin\varphi, \boldsymbol{0}_{n-2})$, $\boldsymbol{e}_\varphi = (-\sin\varphi, \cos\varphi, \boldsymbol{0}_{n-2})$. Note that $\boldsymbol{e}_w \cdot \frac{\partial L}{\partial \boldsymbol{w}} = 0$ is satisfied due to the scale symmetry for the loss.

## A.2 CLASSIFICATION OF STATIONARY POINTS

Using Eq.(21), the loss at the three stationary points is rewritten as

$$\begin{aligned}
L &= -\frac{|\boldsymbol{w}^*|^2}{4\pi}\boldsymbol{v}^{*T}\left(\cos^2\varphi \boldsymbol{I}_{m\times m} + \left(\frac{-(\cos\varphi - 1)^2}{m+1} + 1\right)\boldsymbol{1}_m \boldsymbol{1}_m^T\right)\boldsymbol{v}^* \\
&\quad + \frac{|\boldsymbol{w}^*|^2}{2\pi}\boldsymbol{v}^{*T}\left((\pi - 1)\boldsymbol{I}_{m\times m} + \boldsymbol{1}_m \boldsymbol{1}_m^T\right)\boldsymbol{v}^*.
\end{aligned} \quad (46)$$

Obviously, this shows that $\varphi = 0$ is the global minimum point,

$$L \geq L|_{\varphi=0} = \frac{|\boldsymbol{w}^*|^2}{4\pi}(\pi - 2)\boldsymbol{v}^{*T}\boldsymbol{v}. \tag{47}$$

At $\varphi = \pi$, this becomes

$$
\begin{aligned}
L|_{\varphi=\pi} &= -\frac{|\boldsymbol{w}^*|^2}{4\pi}\boldsymbol{v}^{*T}\left(\boldsymbol{I}_{m\times m} + \left(\frac{-4}{m+1} + 1\right)\mathbf{1}_m\mathbf{1}_m^T\right)\boldsymbol{v}^* \\
&\quad + \frac{|\boldsymbol{w}^*|^2}{4\pi}\boldsymbol{v}^{*T}\left((\pi - 1)\boldsymbol{I}_{m\times m} + \mathbf{1}_m\mathbf{1}_m^T\right)\boldsymbol{v}^* \\
&= \frac{|\boldsymbol{w}^*|^2}{4\pi}\boldsymbol{v}^{*T}\left((\pi - 2)\boldsymbol{I}_{m\times m} + \frac{4}{m+1}\mathbf{1}_m\mathbf{1}_m^T\right)\boldsymbol{v}^* \tag{48}
\end{aligned}
$$

At $\varphi = \bar{\varphi} = \arccos\left[\frac{\left(\boldsymbol{v}^{*T}\mathbf{1}_m\right)^2}{-(m+1)(\boldsymbol{v}^*)^2 + (\boldsymbol{v}^{*T}\mathbf{1}_m)^2}\right]$, this becomes

$$
\begin{aligned}
L|_{\varphi=\bar{\varphi}} &= -\frac{|\boldsymbol{w}^*|^2}{4\pi}\boldsymbol{v}^{*T}\left[\left\{\frac{\left(\boldsymbol{v}^{*T}\mathbf{1}_m\right)^2}{-(m+1)(\boldsymbol{v}^*)^2 + (\boldsymbol{v}^{*T}\mathbf{1}_m)^2}\right\}^2\boldsymbol{I}_{m\times m}\right. \\
&\quad + \left.\left(-(m+1)\left\{\frac{(\boldsymbol{v}^*)^2}{-(m+1)(\boldsymbol{v}^*)^2 + (\boldsymbol{v}^{*T}\mathbf{1}_m)^2}\right\}^2 + 1\right)\mathbf{1}_m\mathbf{1}_m^T\right]\boldsymbol{v}^* \\
&\quad + \frac{|\boldsymbol{w}^*|^2}{4\pi}\boldsymbol{v}^{*T}\left((\pi - 1)\boldsymbol{I}_{m\times m} + \mathbf{1}_m\mathbf{1}_m^T\right)\boldsymbol{v}^* \\
&= -\frac{|\boldsymbol{w}^*|^2}{4\pi}\left\{\frac{\left(\boldsymbol{v}^{*T}\mathbf{1}_m\right)^2(\boldsymbol{v}^*)^2}{-(m+1)(\boldsymbol{v}^*)^2 + (\boldsymbol{v}^{*T}\mathbf{1}_m)^2} + \left(\boldsymbol{v}^{*T}\mathbf{1}_m\right)^2\right\} \\
&\quad + \frac{|\boldsymbol{w}^*|^2}{4\pi}\boldsymbol{v}^{*T}\left((\pi - 1)\boldsymbol{I}_{m\times m} + \mathbf{1}_m\mathbf{1}_m^T\right)\boldsymbol{v}^*. \tag{49}
\end{aligned}
$$

Using these results, we obtain

$$
\begin{aligned}
L|_{\varphi=\bar{\varphi}} &- L|_{\varphi=\pi} \\
&= -\frac{|\boldsymbol{w}^*|^2}{4\pi}\frac{\left(\boldsymbol{v}^{*T}\mathbf{1}_m\right)^2(\boldsymbol{v}^*)^2}{-(m+1)(\boldsymbol{v}^*)^2 + (\boldsymbol{v}^{*T}\mathbf{1}_m)^2} + \frac{|\boldsymbol{w}^*|^2}{4\pi}(\boldsymbol{v}^*)^2 - \frac{|\boldsymbol{w}^*|^2}{4\pi}\left(\boldsymbol{v}^{*T}\mathbf{1}_m\right)^2\frac{4}{m+1} \\
&= \frac{|\boldsymbol{w}^*|^2}{4\pi}\frac{\left\{(m+1)(\boldsymbol{v}^*)^2 - 2\left(\boldsymbol{v}^{*T}\mathbf{1}_m\right)^2\right\}^2}{\left\{(m+1)(\boldsymbol{v}^*)^2 - (\boldsymbol{v}^{*T}\mathbf{1}_m)^2\right\}(m+1)}. \tag{50}
\end{aligned}
$$

Note that because $(m+1)(\boldsymbol{v}^*)^2 \geq 2\left(\boldsymbol{v}^{*T}\mathbf{1}_m\right)^2$ is satisfied to exist the stationary point at $\varphi = \bar{\varphi}$, the loss at $\varphi = \pi$ is smaller than the one at at $\varphi = \bar{\varphi}$,

$$L|_{\boldsymbol{v}=\bar{\boldsymbol{v}},\varphi=\bar{\varphi}} \geq L|_{\boldsymbol{v}=\boldsymbol{v}_\pi,\varphi=\pi}. \tag{51}$$

is satisfied. When $(m+1)(\boldsymbol{v}^*)^2 = 2\left(\boldsymbol{v}^{*T}\mathbf{1}_m\right)^2$, $L|_{\varphi=\bar{\varphi}} = L|_{\varphi=\pi}$, because the middle stationary point at $\varphi = \bar{\varphi}$ is merged into the point at $\varphi = \pi$.

This implies that $\varphi = 0$ is the global minimum, $\varphi = \pi$ is the local minimum, and $\varphi = \bar{\varphi}$ is the saddle point or the local maximum if $(m+1)(\boldsymbol{v}^*)^2 \geq 2\left(\boldsymbol{v}^{*T}\mathbf{1}_m\right)^2$.

To fully clarify whether the stationary points are local minimum, maximum or saddle point, we have calculated the hessian matrix,

$$
\begin{aligned}
\mathcal{H} &= \begin{pmatrix} \frac{\partial^2 L}{\partial \boldsymbol{v}^2} & \frac{\partial^2 L}{\partial \boldsymbol{v}\partial\varphi} \\ \frac{\partial^2 L}{\partial\varphi\partial\boldsymbol{v}} & \frac{\partial^2 L}{\partial\varphi^2} \end{pmatrix} \\
&= \begin{pmatrix} \frac{1}{4}\left(\boldsymbol{I}_{m\times m} + \mathbf{1}_m\mathbf{1}_m^T\right) & \frac{|\boldsymbol{w}^*|}{2\sqrt{2\pi}}\sin\varphi\boldsymbol{v}^* \\ \frac{|\boldsymbol{w}^*|}{2\sqrt{2\pi}}\sin\varphi\boldsymbol{v}^{*T} & \frac{|\boldsymbol{w}^*|}{2\sqrt{2\pi}}\boldsymbol{v}^T\boldsymbol{v}^*\cos\varphi \end{pmatrix}. \tag{52}
\end{aligned}
$$

(i) $\boldsymbol{v} = \bar{\boldsymbol{v}}, \varphi = \bar{\varphi}$:

For $z = (\boldsymbol{x}, y)^T (\boldsymbol{x} \in \mathbb{R}^m, y \in R)$,

$$
\begin{aligned}
z^T \mathcal{H} z &= \frac{1}{4} \boldsymbol{x}^T \left( \boldsymbol{I}_{m \times m} + \boldsymbol{1}_m \boldsymbol{1}_m^T \right) \boldsymbol{x} + \frac{|\boldsymbol{w}^*|}{\sqrt{2\pi}} y \sin \bar{\varphi} \left( \boldsymbol{x}^T \boldsymbol{v}^* \right) \\
&= \frac{1}{4} \left( \boldsymbol{x}^T \boldsymbol{1}_m \right)^2 + \left( \frac{1}{2} \boldsymbol{x} + \frac{|\boldsymbol{w}^*| y \sin \bar{\varphi}}{\sqrt{2\pi}} \boldsymbol{v}^* \right)^2 - \frac{|\boldsymbol{w}^*|^2 y^2 \sin^2 \bar{\varphi} \left( \boldsymbol{v}^* \right)^2}{2\pi},
\end{aligned}
\tag{53}
$$

where we used $\bar{\boldsymbol{v}}^T \boldsymbol{v}^* = 0$. Therefore, depending on $\boldsymbol{x}$ and $y$, the Hessian can be either positive or negative, which means the stationary point is a saddle point.

(ii) $\boldsymbol{v} = \boldsymbol{v}_\pi \equiv \frac{2|\boldsymbol{w}^*|}{\sqrt{2\pi}} \left( -\boldsymbol{I}_{m \times m} + \frac{2}{m+1} \boldsymbol{1}_m \boldsymbol{1}_m^T \right) \boldsymbol{v}^*, \varphi = \pi$

$$
\mathcal{H} = \begin{pmatrix} \frac{1}{4} \left( \boldsymbol{I}_{m \times m} + \boldsymbol{1}_m \boldsymbol{1}_m^T \right) & 0 \\ 0 & -\frac{|\boldsymbol{w}^*|}{2\sqrt{2\pi}} \boldsymbol{v}_\pi^T \boldsymbol{v}^* \end{pmatrix}.
\tag{54}
$$

Note that $\boldsymbol{v}_\pi^T \boldsymbol{v}^* = \frac{2|\boldsymbol{w}^*|}{\sqrt{2\pi}} \boldsymbol{v}^{*T} \left( -\boldsymbol{I}_{m \times m} + \frac{2}{m+1} \boldsymbol{1}_m \boldsymbol{1}_m^T \right) \boldsymbol{v}^*$. If $(m+1) \left( \boldsymbol{v}^* \right)^2 \geq 2 \left( \boldsymbol{v}^{*T} \boldsymbol{1}_m \right)^2$, then $\boldsymbol{v}_\pi^T \boldsymbol{v}^* \leq 0$, so that the matrix becomes a positive definite matrix: For any $\boldsymbol{x} \in \mathbb{R}^m, y \in R$, this satisfies

$$
\frac{1}{4} \boldsymbol{x}^T \left( \boldsymbol{I}_{m \times m} + \boldsymbol{1}_m \boldsymbol{1}_m^T \right) \boldsymbol{x} - \frac{|\boldsymbol{w}^*|}{2\sqrt{2\pi}} \boldsymbol{v}_\pi^T \boldsymbol{v}^* y^2 \geq 0,
\tag{55}
$$

which means the stationary point is the local minimum. On the other hand, if $(m+1) \left( \boldsymbol{v}^* \right)^2 \leq 2 \left( \boldsymbol{v}^{*T} \boldsymbol{1}_m \right)^2$, the matrix becomes either positive or negative, depending on $\boldsymbol{x}$ and $y$, so that this stationary point becomes a saddle point.

(iii) $\boldsymbol{v} = \boldsymbol{v}_0 \equiv \frac{2|\boldsymbol{w}^*|}{\sqrt{2\pi}} \boldsymbol{v}^*, \varphi = 0$

$$
\mathcal{H} = \begin{pmatrix} \frac{1}{4} \left( \boldsymbol{I}_{m \times m} + \boldsymbol{1}_m \boldsymbol{1}_m^T \right) & 0 \\ 0 & \frac{|\boldsymbol{w}^*|}{2\sqrt{2\pi}} \boldsymbol{v}_0^T \boldsymbol{v}^* \end{pmatrix}.
\tag{56}
$$

The Hessian becomes a positive definite matrix: For any $\boldsymbol{x} \in \mathbb{R}^m, y \in R$, this satisfies

$$
\frac{1}{4} \boldsymbol{x}^T \left( \boldsymbol{I}_{m \times m} + \boldsymbol{1}_m \boldsymbol{1}_m^T \right) \boldsymbol{x} + \frac{|\boldsymbol{w}^*|}{2\sqrt{2\pi}} \boldsymbol{v}_0^T \boldsymbol{v}^* y^2 \geq 0.
\tag{57}
$$

### A.3 DERIVATION OF IDENTITY STE GRADIENT

By substituting $\mu'(\boldsymbol{x}) = 1$ into Eq.(23), we obtain

$$
\begin{aligned}
\mathbb{E}_{\boldsymbol{Z}} [\boldsymbol{g}]_i &= \left( \frac{1}{\sqrt{2\pi}} \right)^{mn} \int \left( \prod_{k,l} dZ_{kl} \right) e^{-\frac{1}{2} \sum_{k,l} Z_{kl}^2} \left( \sum_j Z_{ji} v_j \right) \\
&\quad \times \left( \sum_k v_k \sigma \left( \sum_l Z_{kl} w_l \right) - \sum_k v_k^* f_{\text{Relu}} \left( \sum_l Z_{kl} w_l^* \right) \right).
\end{aligned}
\tag{58}
$$

We calculate each term as follows.

#### A.3.1 CALCULATION OF THE FIRST TERM

We calculate the first term as was done in Lemma 9 (and also Lemma 11) in Yin et al. (2019).

To discuss the first term, we evaluate

$$
F_{ijk}^{\text{id}} \equiv \mathbb{E}_{\boldsymbol{Z}} \left[ Z_{ji} \left[ \sigma(\boldsymbol{Z} \boldsymbol{w}) \right]_k \right].
\tag{59}
$$

Using $F_{ijk}^{\mathrm{id}}$, the first term can be expressed as

$$\text{first term} = \sum_{j,k} F_{ijk}^{\mathrm{id}} v_j v_k. \tag{60}$$

(i) $j \neq k$ case:

$$F_{ijk}^{\mathrm{id}} = 0, \tag{61}$$

because of the odd symmetry for $i$-th and $j$-th components in the integrand.

(ii) $j = k$ case:

$$
\begin{aligned}
F_{ijj}^{\mathrm{id}} &= \left(\frac{1}{\sqrt{2\pi}}\right)^n \int \left(\prod_l dZ_{jl}\right) e^{-\frac{1}{2}\sum_l Z_{jl}^2} Z_{ji} \sigma\left(\sum_l Z_{jl} w_l\right) \\
&= \left(\frac{1}{\sqrt{2\pi}}\right)^n \int \left(\prod_l d\tilde{Z}_{jl}\right) e^{-\frac{1}{2}\sum_l \tilde{Z}_{jl}^2} \left(\sum_{m'} \mathcal{O}_{im'}^{j\,T} \tilde{Z}_{jm'}\right) \sigma\left(\tilde{Z}_{j1}\right) \\
&= \left(\frac{1}{\sqrt{2\pi}}\right)^n \int \left(\prod_l d\tilde{Z}_{jl}\right) e^{-\frac{1}{2}\sum_l \tilde{Z}_{jl}^2} \mathcal{O}_{i1}^{j\,T} \tilde{Z}_{j1} \sigma\left(\tilde{Z}_{j1}\right) \\
&= \left(\frac{1}{\sqrt{2\pi}}\right) \int d\tilde{Z}_{j1} e^{-\frac{1}{2}\tilde{Z}_{j1}^2} \mathcal{O}_{i1}^{j\,T} \tilde{Z}_{j1} \sigma\left(\tilde{Z}_{j1}\right) \\
&= \left(\frac{1}{\sqrt{2\pi}}\right) \mathcal{O}_{i1}^{j\,T} = \left(\frac{1}{\sqrt{2\pi}}\right) \mathcal{O}_{1i}^{j} \\
&= \frac{1}{\sqrt{2\pi}} \frac{w_i}{|\boldsymbol{w}|}
\end{aligned} \tag{62}
$$

where we used the orthogonal transformation in the second line:

$$\tilde{Z}_{jl} = \sum_{m'} \mathcal{O}_{lm'}^{j} Z_{jm'} \tag{63}$$

with $\mathcal{O}_{1m'}^{i} = w_{m'}/|\boldsymbol{w}|$.

The first term is reduced to

$$\text{first term} = \frac{1}{\sqrt{2\pi}} \frac{w_i}{|\boldsymbol{w}|} \left(\boldsymbol{v}^T \boldsymbol{v}\right). \tag{64}$$

### A.3.2 CALCULATION OF THE SECOND TERM

To discuss the second term, we evaluate

$$G_{ijk}^{\mathrm{id}} = \mathbb{E}_{\boldsymbol{Z}}\left[Z_{ji} \left[f_{\mathrm{Relu}}(\boldsymbol{Z}\boldsymbol{w}^*)\right]_k\right]. \tag{65}$$

Using $G_{ijk}^{\mathrm{id}}$, the second term can be expressed as

$$\text{second term} = \sum_{j,k} G_{ijk}^{\mathrm{id}} v_j v_k^*. \tag{66}$$

(i) $j \neq k$ case:

$$G_{ijk}^{\mathrm{id}} = 0, \tag{67}$$

because of the odd symmetry.

(ii) $j = k$ case:

$$
\begin{aligned}
G_{ijj}^{\mathrm{id}} &= \left(\frac{1}{\sqrt{2\pi}}\right)^n \int \left(\prod_l dZ_{jl}\right) e^{-\frac{1}{2}\sum_l Z_{jl}^2} Z_{ji} f_{\mathrm{Relu}}\left(\sum_l Z_{jl} w_l^*\right) \\
&= \left(\frac{1}{\sqrt{2\pi}}\right)^n \int \left(\prod_l d\tilde{Z}_{jl}\right) e^{-\frac{1}{2}\sum_l \tilde{Z}_{jl}^2} \left(\sum_{m'} \mathcal{O}_{im'}^{j\,T} \tilde{Z}_{jm'}\right) f_{\mathrm{Relu}}\left(|\boldsymbol{w}^*| \tilde{Z}_{j1}\right) \\
&= \left(\frac{1}{\sqrt{2\pi}}\right)^n \int \left(\prod_l d\tilde{Z}_{jl}\right) e^{-\frac{1}{2}\sum_l \tilde{Z}_{jl}^2} \mathcal{O}_{i1}^{j\,T} \tilde{Z}_{j1} f_{\mathrm{Relu}}\left(|\boldsymbol{w}^*| \tilde{Z}_{j1}\right) \\
&= \left(\frac{1}{\sqrt{2\pi}}\right) \int d\tilde{Z}_{j1} e^{-\frac{1}{2}\tilde{Z}_{j1}^2} \mathcal{O}_{i1}^{j\,T} \tilde{Z}_{j1} f_{\mathrm{Relu}}\left(|\boldsymbol{w}^*| \tilde{Z}_{j1}\right) \\
&= \left(\frac{1}{\sqrt{2\pi}}\right) \int d\tilde{Z}_{j1} e^{-\frac{1}{2}\tilde{Z}_{j1}^2} \mathcal{O}_{i1}^{j\,T} \tilde{Z}_{j1}^2 |\boldsymbol{w}^*| \sigma\left(|\boldsymbol{w}^*| \tilde{Z}_{j1}\right) \\
&= \frac{|\boldsymbol{w}^*|}{2} \mathcal{O}_{i1}^{j\,T} = \frac{|\boldsymbol{w}^*|}{2} \mathcal{O}_{1i}^{j} \\
&= \frac{1}{2} w_i^*
\end{aligned}
\tag{68}
$$

where we have used the orthogonal transformation in the second line:

$$
\tilde{Z}_{jl} = \sum_{m'} \mathcal{O}_{lm'}^{j} Z_{jm'}
\tag{69}
$$

with $\mathcal{O}_{1m'}^{i} = w_{m'}^*/|\boldsymbol{w}^*|$, and also have used

$$
\int_0^\infty dx\, x^2 e^{-(1/2)x^2} = \frac{\sqrt{2\pi}}{2}.
\tag{70}
$$

The second term is written as

$$
\text{second term} = \frac{1}{2} w_i \left(\boldsymbol{v}^T \boldsymbol{v}^*\right).
\tag{71}
$$

### A.4 DERIVATION OF RELU STE GRADIENT

By substituting $\mu'(\boldsymbol{x}) = \sigma(\boldsymbol{x})$ into Eq.(23), we obtain

$$
\begin{aligned}
\mathbb{E}_{\boldsymbol{Z}}\left[\boldsymbol{g}_{\mathrm{relu}}\right]_i &= \left(\frac{1}{\sqrt{2\pi}}\right)^{mn} \int \left(\prod_{k,l} dZ_{kl}\right) e^{-\frac{1}{2}\sum_{k,l} Z_{kl}^2} \left(\sum_j Z_{ji} v_j \sigma\left(\sum_l Z_{jl} w_l\right)\right) \\
&\quad \times \left(\sum_k v_k \sigma\left(\sum_l Z_{kl} w_l\right) - \sum_k v_k^* f_{\mathrm{Relu}}\left(\sum_l Z_{kl} w_l^*\right)\right).
\end{aligned}
\tag{72}
$$

We calculate each term as follows.

#### A.4.1 CALCULATION OF THE FIRST TERM

We calculate the first term as was done in Lemma9 (and also Lemma11) in Yin et al. (2019). We evaluate

$$
F_{ijk}^{\mathrm{relu}} = \mathbb{E}_{\boldsymbol{Z}}\left[Z_{ji} \left[\sigma(\boldsymbol{Z}\boldsymbol{w})\right]_j \left[\sigma(\boldsymbol{Z}\boldsymbol{w})\right]_k\right].
\tag{73}
$$

Using $F_{ijk}^{\mathrm{relu}}$, the first term can be expressed as

$$
\text{first term} = \sum_{j,k} F_{ijk}^{\mathrm{relu}} v_j v_k.
\tag{74}
$$

(i) $j \neq k$ case:

$$
\begin{aligned}
F_{ijk}^{\mathrm{relu}} &= \left(\frac{1}{\sqrt{2\pi}}\right)^{2n} \int \left(\prod_l dZ_{jl} dZ_{kl}\right) e^{-\frac{1}{2}\sum_l Z_{jl}^2 + Z_{kl}^2} Z_{ji} \sigma\left(\sum_l Z_{jl} w_l\right) \sigma\left(\sum_l Z_{kl} w_l\right) \\
&= \left(\frac{1}{\sqrt{2\pi}}\right)^{2n} \int \left(\prod_l d\tilde{Z}_{jl} d\tilde{Z}_{kl}\right) e^{-\frac{1}{2}\sum_l \tilde{Z}_{jl}^2 + \tilde{Z}_{kl}^2} \left(\sum_{m'} \mathcal{O}_{im'}^{j\,T} \tilde{Z}_{jm'}\right) \sigma\left(\tilde{Z}_{j1}\right) \sigma\left(\tilde{Z}_{k1}\right) \\
&= \left(\frac{1}{\sqrt{2\pi}}\right)^2 \int d\tilde{Z}_{j1} d\tilde{Z}_{k1} e^{-\frac{1}{2}\left(\tilde{Z}_{j1}^2 + \tilde{Z}_{k1}^2\right)} \mathcal{O}_{i1}^{j\,T} \tilde{Z}_{j1} \sigma\left(\tilde{Z}_{j1}\right) \sigma\left(\tilde{Z}_{k1}\right) \\
&= \frac{1}{2\sqrt{2\pi}} \mathcal{O}_{i1}^{j\,T} \\
&= \frac{1}{2\sqrt{2\pi}} \frac{w_i}{|\boldsymbol{w}|},
\end{aligned}
\tag{75}
$$

where we used the orthogonal transformation in the second line:

$$
\tilde{Z}_{jl} = \sum_{m'} \mathcal{O}_{lm'}^{j} Z_{jm'}
\tag{76}
$$

with $\mathcal{O}_{1m'}^{i} = w_{m'}/|\boldsymbol{w}|$.

(ii) $j = k$ case:

We get the same expression given in Eq.(62),

$$
\begin{aligned}
F_{ijj}^{\mathrm{relu}} &= \left(\frac{1}{\sqrt{2\pi}}\right)^{n} \int \left(\prod_l dZ_{jl}\right) e^{-\frac{1}{2}\sum_l Z_{jl}^2} Z_{ji} \sigma\left(\sum_l Z_{jl} w_l\right) \\
&= \frac{1}{\sqrt{2\pi}} \frac{w_i}{|\boldsymbol{w}|}.
\end{aligned}
\tag{77}
$$

The first term is written as

$$
\text{first term} = \frac{1}{2\sqrt{2\pi}} \frac{w_i}{|\boldsymbol{w}|} \boldsymbol{v}^T \left(\boldsymbol{I}_{m\times m} + \boldsymbol{1}_m \boldsymbol{1}_m^T\right) \boldsymbol{v}.
\tag{78}
$$

### A.4.2 CALCULATION OF SECOND TERM

To discuss the second term, we evaluate

$$
G_{ijk}^{\mathrm{relu}} = \mathbb{E}_{\boldsymbol{Z}} \left[ Z_{ji} \left[ [\sigma(\boldsymbol{Z}\boldsymbol{w})]_j \, f_{\mathrm{Relu}}(\boldsymbol{Z}\boldsymbol{w}^*) \right]_k \right].
\tag{79}
$$

Using $G_{ijk}^{\mathrm{relu}}$, the second term can be expressed as

$$
\text{second term} = \sum_{j,k} G_{ijk}^{\mathrm{relu}} \boldsymbol{v}_j \boldsymbol{v}_k^*.
\tag{80}
$$

(i) $j \neq k$ case:

$$
\begin{aligned}
G_{ijk}^{\mathrm{relu}} &= \left(\frac{1}{\sqrt{2\pi}}\right)^{2n} \int \left(\prod_l dZ_{jl} dZ_{kl}\right) e^{-\frac{1}{2}\sum_l Z_{jl}^2 + Z_{kl}^2} Z_{ji} \sigma\left(\sum_l Z_{jl} w_l\right) f_{\mathrm{Relu}}\left(\sum_l Z_{kl} w_l^*\right) \\
&= \left(\frac{1}{\sqrt{2\pi}}\right)^{2n} \int \left(\prod_l d\tilde{Z}_{jl} d\tilde{Z}_{kl}\right) e^{-\frac{1}{2}\sum_l \tilde{Z}_{jl}^2 + \tilde{Z}_{kl}^2} \left(\sum_{m'} \mathcal{O}_{im'}^{j\,T} \tilde{Z}_{jm'}\right) \sigma\left(\tilde{Z}_{j1}\right) f_{\mathrm{Relu}}\left(|\boldsymbol{w}^*|\tilde{Z}_{k1}\right) \\
&= \left(\frac{1}{\sqrt{2\pi}}\right)^2 \int d\tilde{Z}_{j1} d\tilde{Z}_{k1} e^{-\frac{1}{2}\left(\tilde{Z}_{j1}^2 + \tilde{Z}_{k1}^2\right)} \mathcal{O}_{i1}^{j\,T} \tilde{Z}_{j1} \sigma\left(\tilde{Z}_{j1}\right) |\boldsymbol{w}^*|\tilde{Z}_{k1} \sigma\left(\tilde{Z}_{k1}\right) \\
&= \frac{|\boldsymbol{w}^*|}{2\pi} \mathcal{O}_{i1}^{j\,T} \\
&= \frac{1}{2\pi} \frac{|\boldsymbol{w}^*|}{|\boldsymbol{w}|} w_i,
\end{aligned}
\tag{81}
$$

where we used the orthogonal transformation in the second line:

$$\tilde{Z}_{jl} = \sum_{m'} \mathcal{O}^j_{lm'} Z_{jm'},$$

$$\tilde{Z}_{kl} = \sum_{m'} \mathcal{O}^k_{lm'} Z_{km'} \tag{82}$$

with $\mathcal{O}^i_{1m'} = w_{m'}/|\boldsymbol{w}|$ and $\mathcal{O}^j_{1m'} = w^*_{m'}/|\boldsymbol{w}^*|$.

(ii) $j = k$ case:

$$
\begin{aligned}
G^{\text{relu}}_{ijj} &= \left(\frac{1}{\sqrt{2\pi}}\right)^m \int \left(\prod_l dZ_{jl}\right) e^{-\frac{1}{2}\sum_l Z^2_{jl}} Z_{ji}\sigma\left(\sum_l Z_{jl}w_l\right) f_{\text{Relu}}\left(\sum_l Z_{jl}w^*_l\right) \\
&= \left(\frac{1}{\sqrt{2\pi}}\right)^m \int \left(\prod_l dZ_{jl}\right) e^{-\frac{1}{2}\sum_l Z^2_{jl}} Z_{ji}\sigma\left(\sum_{l=1}^2 Z_{jl}w_l\right) f_{\text{Relu}}\left(w^* Z_{j1}\right) \\
&= \frac{1}{2\pi}\int \left(\prod_{l=1}^2 dZ_{jl}\right) e^{-\frac{1}{2}\sum_{l=1}^2 Z^2_{jl}} Z_{ji}(\delta_{i1}+\delta_{i2})\sigma\left(\sum_{l=1}^2 Z_{jl}w_l\right) f_{\text{Relu}}\left(w^* Z_{j1}\right) \\
&= \frac{1}{2\pi}\int dZ d\theta Z e^{-\frac{1}{2}Z^2}\left(Z\cos\theta\delta_{i1}+Z\sin\theta\delta_{i2}\right)\sigma\left(Zw\cos(\theta-\varphi)\right) f_{\text{Relu}}\left(w^* Z\cos\theta\right) \\
&= \frac{1}{2\pi}\int dZ d\theta Z e^{-\frac{1}{2}Z^2}\left(Z\cos\theta\delta_{i1}+Z\sin\theta\delta_{i2}\right)\sigma\left(Zw\cos(\theta-\varphi)\right) w^* Z\cos\theta\sigma\left(\cos\theta\right)
\end{aligned}
\tag{83}
$$

$$
\begin{aligned}
&= \frac{w^*}{2\pi}\int_0^\infty dZ Z^3 e^{-\frac{1}{2}Z^2}\int_{-\pi/2+\varphi}^{\pi/2} d\theta\left(\cos^2\theta\delta_{i1}+\cos\theta\sin\theta\delta_{i2}\right) \\
&= \frac{w^*}{\pi}\left(\left(\frac{\sin 2\varphi}{4}+\frac{1}{2}(\pi-\varphi)\right)\delta_{i1}+\frac{1-\cos 2\varphi}{4}\delta_{i2}\right) \\
&= \frac{w^*}{\pi}\left(\frac{1}{2}(\pi-\varphi)\delta_{i1}+\frac{\sin\varphi}{2}\left(\cos\varphi\delta_{i1}+\sin\varphi\delta_{i2}\right)\right)
\end{aligned}
\tag{84}
$$

where we have chosen

$$
\begin{aligned}
\boldsymbol{w}^* &= (w^*, \boldsymbol{0}_{n-1}), \\
\boldsymbol{w} &= (w_1, w_2, \boldsymbol{0}_{n-2})
\end{aligned}
\tag{85}
$$

with $w^* > 0$ and $\varphi$ is defined as the angle between $\boldsymbol{w}^*$ and $\boldsymbol{w}$. We also have used the formula,

$$
\int_0^\infty dx x^3 e^{-(1/2)x^2} = 2,
$$

$$
\int d\theta \cos^2\theta = \frac{\sin 2\theta}{4} + \frac{1}{2}\theta,
$$

$$
\int d\theta \cos\theta\sin\theta = -\frac{1}{4}\cos 2\theta. \tag{86}
$$

Note that

$$
\begin{aligned}
\frac{\boldsymbol{w}^*_i}{|\boldsymbol{w}^*|} &= \delta_{i1}, \\
\frac{\boldsymbol{w}_i}{|\boldsymbol{w}|} &= \cos\varphi\delta_{i1}+\sin\varphi\delta_{i2}.
\end{aligned}
\tag{87}
$$

This reduces to

$$
G^{\text{relu}}_{ijj} = \frac{1}{\pi}\left(\frac{1}{2}(\pi-\varphi)\boldsymbol{w}^*_i+\frac{\sin\varphi}{2}\frac{|\boldsymbol{w}^*|}{|\boldsymbol{w}|}\boldsymbol{w}_i\right) \tag{88}
$$

In summary, the second term is written as

$$
\begin{aligned}
\text{second term} &= \frac{1}{2\pi}\frac{|\boldsymbol{w}^*|}{|\boldsymbol{w}|}\boldsymbol{w}_i\left[\boldsymbol{v}^T\left(-\boldsymbol{I}_{m\times m}+\boldsymbol{1}_m\boldsymbol{1}_m^T\right)\boldsymbol{v}^*\right] \\
&\quad + \left((\pi-\varphi)\boldsymbol{w}^*_i+\frac{|\boldsymbol{w}^*|\sin\varphi}{|\boldsymbol{w}|}\boldsymbol{w}_i\right)\frac{(\boldsymbol{v}^T\boldsymbol{v}^*)}{2\pi}.
\end{aligned}
\tag{89}
$$

## A.5 DERIVATION OF CLIPPED RELU STE GRADIENT

By substituting $\mu'(x) = \sigma(x)\sigma(r-x)$ into Eq.(23), we obtain

$$
\mathbb{E}_{\boldsymbol{Z}}\left[\boldsymbol{g}_{\mathrm{crelu}}\right]_i = \left(\frac{1}{\sqrt{2\pi}}\right)^{mn} \int \left(\prod_{k,l} dZ_{kl}\right) e^{-\frac{1}{2}\sum_{k,l} Z_{kl}^2}
$$
$$
\times \left(\sum_j Z_{ji} v_j \sigma\left(\sum_l Z_{jl} w_l\right)\sigma\left(r - \sum_l Z_{jl} w_l\right)\right)
$$
$$
\times \left(\sum_k v_k \sigma\left(\sum_l Z_{kl} w_l\right) - \sum_k v_k^* f_{\mathrm{Relu}}\left(\sum_l Z_{kl} w_l^*\right)\right). \tag{90}
$$

We calculate each term as follows.

### A.5.1 CALCULATION OF THE FIRST TERM

To discuss the first term, we evaluate

$$
F_{ijk}^{\mathrm{crelu}} = \mathbb{E}_{\boldsymbol{Z}}\left[Z_{ji}\left[\sigma(\boldsymbol{Zw}) \odot \sigma(r - \boldsymbol{Zw})\right]_j \left[\sigma(\boldsymbol{Zw})\right]_k\right]. \tag{91}
$$

Using $F_{ijk}^{\mathrm{crelu}}$, the first term can be expressed as

$$
\text{first term} = \sum_{j,k} F_{ijk}^{\mathrm{crelu}} v_j v_k. \tag{92}
$$

(i) $j \neq k$ case:

$$
F_{ijk}^{\mathrm{crelu}} = \left(\frac{1}{\sqrt{2\pi}}\right)^{2n} \int \left(\prod_l dZ_{jl} dZ_{kl}\right) e^{-\frac{1}{2}\sum_l Z_{jl}^2 + Z_{kl}^2} Z_{ji}
$$
$$
\times \sigma\left(\sum_l Z_{jl} w_l\right)\sigma\left(r - \sum_l Z_{jl} w_l\right)\sigma\left(\sum_l Z_{kl} w_l\right)
$$
$$
= \left(\frac{1}{\sqrt{2\pi}}\right)^{2n} \int \left(\prod_l d\tilde{Z}_{jl} d\tilde{Z}_{kl}\right) e^{-\frac{1}{2}\sum_l \tilde{Z}_{jl}^2 + \tilde{Z}_{kl}^2} \left(\sum_{m'} \mathcal{O}_{im'}^{j\ T} \tilde{Z}_{jm'}\right)
$$
$$
\times \sigma\left(\tilde{Z}_{j1}\right)\sigma\left(r - |\boldsymbol{w}|\tilde{Z}_{j1}\right)\sigma\left(\tilde{Z}_{k1}\right)
$$
$$
= \left(\frac{1}{\sqrt{2\pi}}\right)^2 \int d\tilde{Z}_{j1} d\tilde{Z}_{k1} e^{-\frac{1}{2}\left(\tilde{Z}_{j1}^2 + \tilde{Z}_{k1}^2\right)} \mathcal{O}_{i1}^{j\ T} \tilde{Z}_{j1}\sigma\left(\tilde{Z}_{j1}\right)\sigma\left(r - |\boldsymbol{w}|\tilde{Z}_{j1}\right)\sigma\left(\tilde{Z}_{k1}\right)
$$
$$
= \frac{1}{2\sqrt{2\pi}} \int_0^{r/|\boldsymbol{w}|} d\tilde{Z}_{j1} e^{-\frac{1}{2}\tilde{Z}_{j1}^2} \mathcal{O}_{i1}^{j\ T} \tilde{Z}_{j1}
$$
$$
= \frac{1}{2\sqrt{2\pi}} \mathcal{O}_{i1}^{j\ T}\left(1 - e^{-\frac{1}{2}\left(\frac{r}{|\boldsymbol{w}|}\right)^2}\right)
$$
$$
= \frac{1}{2\sqrt{2\pi}}\left(1 - e^{-\frac{1}{2}\left(\frac{r}{|\boldsymbol{w}|}\right)^2}\right)\frac{w_i}{|\boldsymbol{w}|}, \tag{93}
$$

where we used the orthogonal transformation in the second line:

$$
\tilde{Z}_{jl} = \sum_{m'} \mathcal{O}_{lm'}^j Z_{jm'} \tag{94}
$$

with $\mathcal{O}_{1m'}^i = w_{m'}/|\boldsymbol{w}|$.

(ii) $j = k$ case:

$$
\begin{aligned}
F_{ijj}^{\text{crelu}} &= \left(\frac{1}{\sqrt{2\pi}}\right)^n \int \left(\prod_l dZ_{jl}\right) e^{-\frac{1}{2}\sum_l Z_{jl}^2} Z_{ji}\sigma\left(\sum_l Z_{jl}w_l\right)\sigma\left(r - \sum_l Z_{jl}w_l\right) \\
&= \left(\frac{1}{\sqrt{2\pi}}\right)^n \int \left(\prod_l d\tilde{Z}_{jl}\right) e^{-\frac{1}{2}\sum_l \tilde{Z}_{jl}^2} \left(\sum_{m'} \mathcal{O}_{im'}^{j\,T}\tilde{Z}_{jm'}\right)\sigma\left(\tilde{Z}_{j1}\right)\sigma\left(r - |\boldsymbol{w}|\tilde{Z}_{j1}\right) \\
&= \frac{1}{\sqrt{2\pi}} \int d\tilde{Z}_{j1} e^{-\frac{1}{2}\tilde{Z}_{j1}^2} \mathcal{O}_{i1}^{j\,T}\tilde{Z}_{j1}\sigma\left(\tilde{Z}_{j1}\right)\sigma\left(r - |\boldsymbol{w}|\tilde{Z}_{j1}\right) \\
&= \frac{1}{\sqrt{2\pi}} \int_0^{r/|\boldsymbol{w}|} d\tilde{Z}_{j1} e^{-\frac{1}{2}\tilde{Z}_{j1}^2} \mathcal{O}_{i1}^{j\,T}\tilde{Z}_{j1} \\
&= \frac{1}{\sqrt{2\pi}} \mathcal{O}_{i1}^{j\,T}\left(1 - e^{-\frac{1}{2}\left(\frac{r}{|\boldsymbol{w}|}\right)^2}\right) \\
&= \frac{1}{\sqrt{2\pi}}\left(1 - e^{-\frac{1}{2}\left(\frac{r}{|\boldsymbol{w}|}\right)^2}\right)\frac{w_i}{|\boldsymbol{w}|},
\end{aligned}
\tag{95}
$$

In summary, the first term is written as

$$
\text{first term} = \frac{1}{2\sqrt{2\pi}}\frac{w_i}{|\boldsymbol{w}|}\left(1 - e^{-\frac{1}{2}\left(\frac{r}{|\boldsymbol{w}|}\right)^2}\right)\boldsymbol{v}^T\left(\boldsymbol{I}_{m\times m} + \boldsymbol{1}_m\boldsymbol{1}_m^T\right)\boldsymbol{v}.
\tag{96}
$$

### A.5.2 Calculation of the second term

To discuss the second term, we evaluate

$$
G_{ijk}^{\text{crelu}} = \mathbb{E}_{\boldsymbol{Z}}\left[Z_{ji}\left[\left[\sigma(\boldsymbol{Z}\boldsymbol{w}) \odot \sigma(r - \boldsymbol{Z}\boldsymbol{w})\right]_j f_{\text{Relu}}(\boldsymbol{Z}\boldsymbol{w}^*)\right]_k\right].
\tag{97}
$$

Using $G_{ijk}^{\text{crelu}}$, the second term can be expressed as

$$
\text{second term} = \sum_{j,k} G_{ijk}^{\text{crelu}}\boldsymbol{v}_j\boldsymbol{v}_k^*.
\tag{98}
$$

(i) $j \neq k$ case:

$$
\begin{aligned}
G_{ijk}^{\text{crelu}} &= \left(\frac{1}{\sqrt{2\pi}}\right)^{2n} \int \left(\prod_l dZ_{jl}dZ_{kl}\right) e^{-\frac{1}{2}\sum_l Z_{jl}^2+Z_{kl}^2} Z_{ji} \\
&\quad \times \sigma\left(\sum_l Z_{jl}w_l\right)\sigma\left(r - \sum_l Z_{jl}w_l\right) f_{\text{Relu}}\left(\sum_l Z_{kl}w_l^*\right) \\
&= \left(\frac{1}{\sqrt{2\pi}}\right)^{2n} \int \left(\prod_l d\tilde{Z}_{jl}d\tilde{Z}_{kl}\right) e^{-\frac{1}{2}\sum_l \tilde{Z}_{jl}^2+\tilde{Z}_{kl}^2} \left(\sum_{m'} \mathcal{O}_{im'}^{j\,T}\tilde{Z}_{jm'}\right) \\
&\quad \times \sigma\left(\tilde{Z}_{j1}\right)\sigma\left(r - |\boldsymbol{w}|\tilde{Z}_{j1}\right) f_{\text{Relu}}\left(|\boldsymbol{w}^*|\tilde{Z}_{k1}\right) \\
&= \left(\frac{1}{\sqrt{2\pi}}\right)^2 \int d\tilde{Z}_{j1}d\tilde{Z}_{k1} e^{-\frac{1}{2}\left(\tilde{Z}_{j1}^2+\tilde{Z}_{k1}^2\right)} \mathcal{O}_{i1}^{j\,T}\tilde{Z}_{j1} \\
&\quad \times \sigma\left(\tilde{Z}_{j1}\right)\sigma\left(r - |\boldsymbol{w}|\tilde{Z}_{j1}\right)|\boldsymbol{w}^*|\tilde{Z}_{k1}\sigma\left(\tilde{Z}_{k1}\right) \\
&= \left(\frac{1}{\sqrt{2\pi}}\right)^2 \int d\tilde{Z}_{j1} e^{-\frac{1}{2}\tilde{Z}_{j1}^2} \mathcal{O}_{i1}^{j\,T}\tilde{Z}_{j1}\sigma\left(\tilde{Z}_{j1}\right)\sigma\left(r - |\boldsymbol{w}|\tilde{Z}_{j1}\right)|\boldsymbol{w}^*| \\
&= \left(\frac{1}{\sqrt{2\pi}}\right)^2 \int_0^{r/|\boldsymbol{w}|} d\tilde{Z}_{j1} e^{-\frac{1}{2}\tilde{Z}_{j1}^2} \mathcal{O}_{i1}^{j\,T}\tilde{Z}_{j1}|\boldsymbol{w}^*| \\
&= \frac{|\boldsymbol{w}^*|}{2\pi}\mathcal{O}_{i1}^{j\,T}\left(1 - e^{-\frac{1}{2}\left(\frac{r}{|\boldsymbol{w}|}\right)^2}\right) \\
&= \frac{|\boldsymbol{w}^*|}{2\pi}\left(1 - e^{-\frac{1}{2}\left(\frac{r}{|\boldsymbol{w}|}\right)^2}\right)\frac{w_i}{|\boldsymbol{w}|},
\end{aligned}
\tag{99}
$$

where we used the orthogonal transformation in the second line:

$$\tilde{Z}_{il} = \sum_{m'} \mathcal{O}^i_{lm'} Z_{im'},$$

$$\tilde{Z}_{kl} = \sum_{m'} \mathcal{O}^k_{lm'} Z_{km'} \tag{100}$$

with $\mathcal{O}^i_{1m'} = w_{m'}/|\boldsymbol{w}|$ and $\mathcal{O}^j_{1m'} = w^*_{m'}/|\boldsymbol{w}^*|$.

(ii) $j = k$ case:

$$G^{\text{crelu}}_{ijj} = \left(\frac{1}{\sqrt{2\pi}}\right)^m \int \left(\prod_l dZ_{jl}\right) e^{-\frac{1}{2}\sum_l Z^2_{jl}} Z_{ji}$$

$$\times \sigma\left(\sum_l Z_{jl}w_l\right) \sigma\left(r - \sum_l Z_{jl}w_l\right) f_{\text{Relu}}\left(\sum_l Z_{jl}w^*_l\right)$$

$$= \left(\frac{1}{\sqrt{2\pi}}\right)^m \int \left(\prod_l dZ_{jl}\right) e^{-\frac{1}{2}\sum_l Z^2_{jl}} Z_{ji}$$

$$\times \sigma\left(\sum_{l=1}^2 Z_{jl}w_l\right) \sigma\left(r - \sum_{l=1}^2 Z_{jl}w_l\right) f_{\text{Relu}}\left(w^* Z_{j1}\right)$$

$$= \frac{1}{2\pi} \int \left(\prod_{l=1}^2 dZ_{jl}\right) e^{-\frac{1}{2}\sum_{l=1}^2 Z^2_{jl}} Z_{ji} \left(\delta_{i1} + \delta_{i2}\right)$$

$$\times \sigma\left(\sum_{l=1}^2 Z_{jl}w_l\right) \sigma\left(r - \sum_{l=1}^2 Z_{jl}w_l\right) f_{\text{Relu}}\left(w^* Z_{j1}\right)$$

$$= \frac{1}{2\pi} \int dZ d\theta Z e^{-\frac{1}{2}Z^2} \left(Z \cos\theta \delta_{i1} + Z \sin\theta \delta_{i2}\right)$$

$$\times \sigma\left(Zw \cos(\theta - \varphi)\right) \sigma\left(r - \sum_{l=1}^2 Z_{jl}w_l\right) f_{\text{Relu}}\left(w^* Z \cos\theta\right)$$

$$= \frac{1}{2\pi} \int dZ d\theta Z e^{-\frac{1}{2}Z^2} \left(Z \cos\theta \delta_{i1} + Z \sin\theta \delta_{i2}\right)$$

$$\times \sigma\left(Zw \cos(\theta - \varphi)\right) \sigma\left(r - Zw \cos(\theta - \varphi)\right) w^* Z \cos\theta \sigma\left(\cos\theta\right) \tag{101}$$

$$= \frac{w^*}{2\pi} \int_{-\pi/2+\varphi}^{\pi/2} d\theta \left(\cos^2\theta \delta_{i1} + \cos\theta \sin\theta \delta_{i2}\right) \int_0^{\frac{r}{w \cos(\theta - \varphi)}} dZ Z^3 e^{-\frac{1}{2}Z^2}$$

$$= \frac{w^*}{2\pi} \int_{-\pi/2+\varphi}^{\pi/2} d\theta \left(\cos^2\theta \delta_{i1} + \cos\theta \sin\theta \delta_{i2}\right)$$

$$\times \left\{ 2 - \left(\left(\frac{r}{w \cos(\theta - \varphi)}\right)^2 + 2\right) e^{-\frac{1}{2}\left(\frac{r}{w \cos(\theta - \varphi)}\right)^2} \right\} \tag{102}$$

where we have chosen

$$\boldsymbol{w}^* = (w^*, \boldsymbol{0}_{n-1}),$$

$$\boldsymbol{w} = (w_1, w_2, \boldsymbol{0}_{n-2}) \tag{103}$$

with $w^* > 0$ and $\varphi$ is defined as the angle between $\boldsymbol{w}^*$ and $\boldsymbol{w}$. We also have used the formula,

$$\int_0^t dx x^3 e^{-\frac{1}{2}x^2} = 2 - (t^2 + 2)e^{-\frac{1}{2}t^2}. \tag{104}$$

Note that

$$\frac{\boldsymbol{w}^*_i}{|\boldsymbol{w}^*|} = \delta_{i1},$$

$$\frac{\boldsymbol{w}_i}{|\boldsymbol{w}|} = \cos\varphi \delta_{i1} + \sin\varphi \delta_{i2}. \tag{105}$$

Using $C(w, \varphi), S(w, \varphi)$ defined as

$$C(w, \varphi) \equiv \frac{w^*}{2\pi} \int_{-\pi/2+\varphi}^{\pi/2} d\theta \cos^2 \theta \left\{ 2 - \left( \left( \frac{r}{w \cos(\theta - \varphi)} \right)^2 + 2 \right) e^{-\frac{1}{2} \left( \frac{r}{w \cos(\theta - \varphi)} \right)^2} \right\},$$

$$S(w, \varphi) \equiv \frac{w^*}{2\pi} \int_{-\pi/2+\varphi}^{\pi/2} d\theta \sin \theta \cos \theta \left\{ 2 - \left( \left( \frac{r}{w \cos(\theta - \varphi)} \right)^2 + 2 \right) e^{-\frac{1}{2} \left( \frac{r}{w \cos(\theta - \varphi)} \right)^2} \right\},$$

$$(106)$$

If $\varphi \neq 0, \pi$, $G_{ijj}^{\text{crelu}}$ is formally written as

$$G_{ijj}^{\text{crelu}} = C(w, \varphi) \frac{\boldsymbol{w}_i^*}{|\boldsymbol{w}^*|} + S(w, \varphi) \left( \frac{1}{\sin \varphi} \frac{\boldsymbol{w}_i}{|\boldsymbol{w}|} - \cot \varphi \frac{\boldsymbol{w}_i^*}{|\boldsymbol{w}^*|} \right) \tag{107}$$

If $\varphi = 0$ or $\pi$, $G_{ijj}^{\text{crelu}}$ is formally written as

$$G_{ijj}^{\text{crelu}} = C(w, 0) \frac{\boldsymbol{w}_i^*}{|\boldsymbol{w}^*|} \quad for \ \varphi = 0, \tag{108}$$

$$G_{ijj}^{\text{crelu}} = 0 \quad for \ \varphi = \pi \tag{109}$$

where we used $S(w, 0) = S(w, \pi) = C(w, \pi) = 0$ and

$$C(w, 0) = \frac{w^*}{2\pi} \int_{-\pi/2}^{\pi/2} d\theta \cos^2 \theta \left\{ 2 - \left( \left( \frac{r}{w \cos \theta} \right)^2 + 2 \right) e^{-\frac{1}{2} \left( \frac{r}{w \cos \theta} \right)^2} \right\}$$

$$= \frac{w^*}{2\pi} \left( \pi - \frac{r}{w} e^{-\frac{1}{2} \left( \frac{r}{w} \right)^2} \sqrt{2\pi} - \pi \text{erfc} \left( \frac{r}{\sqrt{2}w} \right) \right) \tag{110}$$

with the following formula

$$\int_{-\pi/2}^{\pi/2} d\theta e^{-\frac{a^2}{2 \cos^2 \theta}} = \pi \text{erfc} \left( \frac{a}{\sqrt{2}} \right), \tag{111}$$

$$\int_{-\pi/2}^{\pi/2} d\theta \cos^2 \theta e^{-\frac{a^2}{2 \cos^2 \theta}} = \frac{1}{2} \left( a e^{-\frac{a^2}{2}} \sqrt{2\pi} + \left( 1 - a^2 \right) \pi \text{erfc} \left( \frac{a}{\sqrt{2}} \right) \right). \tag{112}$$

In summary, the second term is written as

$$\text{second term} = \frac{1}{2\pi} \frac{|\boldsymbol{w}^*|}{|\boldsymbol{w}|} \left( 1 - e^{-\frac{1}{2} \left( \frac{r}{|\boldsymbol{w}|} \right)^2} \right) \boldsymbol{w}_i \left[ \boldsymbol{v}^T \left( -\boldsymbol{I}_{m \times m} + \boldsymbol{1}_m \boldsymbol{1}_m^T \right) \boldsymbol{v}^* \right]$$

$$+ \left\{ C(w, \varphi) \frac{\boldsymbol{w}_i^*}{|\boldsymbol{w}^*|} + S(w, \varphi) \left( \frac{1}{\sin \varphi} \frac{\boldsymbol{w}_i}{|\boldsymbol{w}|} - \cot \varphi \frac{\boldsymbol{w}_i^*}{|\boldsymbol{w}^*|} \right) \right\} \left( \boldsymbol{v}^T \boldsymbol{v}^* \right). \tag{113}$$

