# OpenReview forum: "Proper Straight-Through Estimator: Breaking symmetry promotes convergence to true minimum"
_ICLR.cc/2022/Conference — ICLR 2022 Submitted_

### Official Review · Reviewer_1Ehv · 2021-11-02

**Correctness:** 3
**Technical Novelty And Significance:** 2
**Empirical Novelty And Significance:** 1
**Recommendation:** 3
**Confidence:** 4

**Main Review:**

The paper contains little novelty compared to Yin et al. (2019)
Also, it is not well written. The claims are weak and it is not clear if they are specific to the specific setting chosen.
There is no numerical validation.

- Claims such as
"""
Considering the scale invariance of Relu STE gradient, this relation implies that the
Relu gradient is approximately embedded in the cRelu STE gradient, so that if the back-
propagation using Relu STE converges to a point, the back-propagation using cRelu STE
can also converge to that point.
"""
are not entirely clear nor sufficiently motivated.


- The main claim reads
"""
In other words, because of the breaking of the scale symmetry, the cRelu STE may pick up one of local minima,
which degenerate in all scales, while because of keeping the scale symmetry, identity STE and Relu
STE may not. Therefore, we conclude that the back-propagation using cRelu STE is the most likely
to achieve the true minimum in the three STEs.
"""
In other words, the authors claim that some minima of the population loss of the original model *may* also be minima of the surrogate model with cReLU STE if the weights are appropriately rescaled. This claim has only the weak analytical support
given by the analysis of the Gaussian input case of the next Section 3. No numerical support is provided for
convergence of the training dynamics in this toy model, let alone deeper networks and structured datasets.

- In Section 3, the authors show that the cReLU STE has an additional stationary point (the global minimum) compared to the
ReLU STE. This is an interesting result, and also somehow natural given that a non-invariant gradient has more degree-of-freedoms
for satisfying the stationary point condition. The single setting explored, the one with the ReLU teacher, doesn't seem enough to support general claims though. Can the author check analytically a few other cases, e.g. tanh activation. Or at least experimentally (possibly also
in deeper networks)?

- Reviewing the paper was made more complicated by the poor writing,
starting from the very first sentence of the abstract  "In the quantized network, its gradient shows either vanishing or diverging."


**Summary Of The Paper:**

The paper investigates analytically the scale invariance properties of
different straight-through estimators used in training neural networks with binary activation and weights.

The main claim is that the cReLU STE, among the 3 considered, is the one that is more likely to give zero gradients in the minima of the population loss. This is (weakly) supported by theoretical analysis on a little variation of the shallow setting analyzed in Yin et al. (2019), where the teacher now contains ReLU activations.


**Summary Of The Review:**

While there is some novelty to the paper, I don't think the importance of the results and the novelty level
are high enough for the ICLR standards.
Moreover, there is a complete lack of experimental validation and some poor writing.
I cannot recommend the manuscript for publication on ICLR.

---

> ### Author Response · Authors · 2021-11-23
> **Response to the Reviewer 1Ehv**
>
> We would like to thank the reviewer for providing the helpful feedback and constructive suggestions. We revised the manuscript according to reviewer’s suggestions. Our replies to your comments are given below.
>
> 	・Claims such as """ Considering the scale invariance of Relu STE gradient, this relation implies that the Relu gradient is approximately embedded in the cRelu STE gradient, so that if the back- propagation using Relu STE converges to a point, the back-propagation using cRelu STE can also converge to that point. """ are not entirely clear nor sufficiently motivated.
>
> Thank you very much for pointing out it. We have removed the the discussion, and changed the sentences to the detailed explanation about Eq.(14) in red as follows:
> If we take $kr$ with fixed $r$ large enough to cover most of the distribution $P(Z)$, the left-hand side is identical with the Relu STE gradient. Here we define the spread of the distribution as $\rho$. When we redefine $w/k$ as $w$ on the right hand, the cRelu STE gradient for $|w| < r/\rho$ is approximately identical with the Relu STE gradient,
>  ~~~~~~~~~~~~~~~~~~~~~~~~~~~~~~~~~~Eq.(14)
>
>   This result is intuitively obvious: when $w$ is small enough to keep most of the pre-activation values in the clipped range, the cRelu gradient and the Relu gradient are approximately identical.
>
>
>
> 	・The main claim reads """ In other words, because of the breaking of the scale symmetry, the cRelu STE may pick up one of local minima, which degenerate in all scales, while because of keeping the scale symmetry, 	identity STE and Relu STE may not. Therefore, we conclude that the back-propagation using cRelu STE is the most likely to achieve the true minimum in the three STEs. """ In other words, the authors claim that some 	minima of the population loss of the original model may also be minima of the surrogate model with cReLU STE if the weights are appropriately rescaled. This claim has only the weak analytical support given by the 	analysis of the Gaussian input case of the next Section 3. No numerical support is provided for convergence of the training dynamics in this toy model, let alone deeper networks and structured datasets.
> 	・In Section 3, the authors show that the cReLU STE has an additional stationary point (the global minimum) compared to the ReLU STE. This is an interesting result, and also somehow natural given that a non-invariant 	gradient has more degree-of-freedoms for satisfying the stationary point condition. The single setting explored, the one with the ReLU teacher, doesn't seem enough to support general claims though. Can the author 	check analytically a few other cases, e.g. tanh activation. Or at least experimentally (possibly also in deeper networks)?
>
> Following your comments and suggestions, to compensate for the theoretical support of our conjecture, we have numerically tested the Gausssian mixture input with various mean values using teacher networks with tanh-type, sin-type as well as Relu-type. For all setup we examined, we have found that cRelu STE is similar to the population gradient, while id/Relu STEs shows completely different behavior: The result with ten Gaussian mixture input with random mean value and tanh-type teacher network is shown in Fig.1. The behaviors are the same as what we discussed/conjectured theoretically.
>
> 	・Reviewing the paper was made more complicated by the poor writing, starting from the very first sentence of the abstract "In the quantized network, its gradient shows either vanishing or diverging."
>
> Thank you very much for the comment. We have modified and sophisticated our manuscript.

---

### Official Review · Reviewer_tphS · 2021-11-02

**Correctness:** 4
**Technical Novelty And Significance:** 2
**Empirical Novelty And Significance:** Not applicable
**Recommendation:** 3
**Confidence:** 5

**Main Review:**

**Strengths:**
The authors propose a clear, concise and closed form analysis of the true gradient (i.e. the gradient of the expected loss) and its stationary points. This is possible due to the simplicity of the model and the assumed data distribution. The same is done for each of the three STE variants. This proves that the first two STE variants are biased in general. Moreover, the authors prove in particular that none of the global optimisers of the expected loss is a stationary point for these STE variants. (Notice that the globally optimal weights of the hidden layer form a ray due to scale invariance). On the other hand, they show that at least one of these global optimisers is also a stationary point for the third STE variant.

**Weaknesses**
The same model has been studied in the previous work Yin et al. (2019) for a different loss. It has been shown there that the first STE variant is biased, while the second and third where found to be unbiased in this case. The loss studied there was the MSE loss w.r.t. a ground truth model with the same architecture. The loss considered in the submitted manuscript is the MSE loss w.r.t. a ground truth model that uses ReLU activations in the hidden layer. The submitted manuscript thus also rules out the second STE variant as biased.

The authors cite the previous work Shekhovtsov & Yanush (2020) which derives STEs as a linearisation of an unbiased gradient estimator with low variance. In particular, it is shown there that the properly derived ST estimator is unbiased for models studied in the submitted manuscript, albeit for a different type of loss. It is clearly seen that their derived unbiased ST estimator breaks the scale invariance. I am pretty sure that the derivation given there can be generalised for the model & loss combination studied in the submitted manuscript.

**Summary Of The Paper:**

The paper considers a network with one hidden layer with binary valued neurons sharing the same weight vector but having non-overlapping receptive fields. The authors analyse the stationary points of a particular MSE loss when a straight through estimator (STE) is used for the loss gradient. They analyse three variants for the STE: (1) derivative of the identity function, (2) derivative of ReLU and (3) derivative of a piecewise linear squashing function. Assuming a simple independent distribution over the inputs and a particular loss, the authors show that the first two variants are biased and only the third STE variant can be unbiased. The authors conjecture that the reason for this is that the third STE variant breaks the scaling symmetry w.r.t. the weights of the first layer.

**Summary Of The Review:**

The paper provides a convincing in-depth analysis of STE properties for a particular simple combination of model and data distribution. However, the novelty is limited in view of the previous work Yin et al. (2019). Its significance is restricted due to the very specific assumptions on the model, data distribution and loss.

---

> ### Author Response · Authors · 2021-11-23
> **Response to the Reviewer tphS**
>
> We would like to thank the reviewer for providing the helpful feedback. Our replies to weak points you mentioned are given below.
>
> 	The same model has been studied in the previous work Yin et al. (2019) for a different loss. It has been shown there that the first STE variant is biased, while the second and third where found to be unbiased in this case. 	The loss studied there was the MSE loss w.r.t. a ground truth model with the same architecture. The loss considered in the submitted manuscript is the MSE loss w.r.t. a ground truth model that uses ReLU activations in 	the hidden layer. The submitted manuscript thus also rules out the second STE variant as biased.
>
> Thank you very much for the comment. Our purpose is not only to study the model by Yin et al. (2019) with a different loss, but also clarify the reason why cRelu STE empirically shows the best performance in the three STEs as discussed in Sec.2: The cRelu breaks the symmetry of the network. The STE with breaking symmetry of the network shows better performance than the ones without breaking the symmetry. Moreover, by generalize the discussion in Sec.2, we conjecture that if you consider some network with more symmetries, and use an STE that breaks all of those symmetries instead of cRelu STE, it is more likely to achieve the minimum than cRelu STE. Therefore, to confirm our claim, we have studied  the model by Yin et al. with a different loss. To the best of my knowledge, there are no previous works that point out the relation between quantized networks and STE based on symmetry.
>
> 	The authors cite the previous work Shekhovtsov & Yanush (2020) which derives STEs as a linearisation of an unbiased gradient estimator with low variance. In particular, it is shown there that the properly derived ST 	estimator is unbiased for models studied in the submitted manuscript, albeit for a different type of loss. It is clearly seen that their derived unbiased ST estimator breaks the scale invariance. I am pretty sure that the 	derivation given there can be generalised for the model & loss combination studied in the submitted manuscript.
>
> In our opinion, Shekhovtsov & Yanush (2020) adds the noise before the binary activation, so that the network itself do not have the scale symmetry. Our claim is applied only to the network with some symmetries, saying that good STE breaks the symmetry of the network. (Just to be clear here, if we add the bias term “b” to our network, this seems to be the similar structure of the model by Shekhovtsov & Yanush (2020), but the network has the scale symmetry for the transformation w -> k * w and b -> k * b Noise is not allowed for the same transformation because it changes the probability distribution.)
>
>
> 	… Its significance is restricted due to the very specific assumptions on the model, data distribution and loss.
>
> Thank you for pointing out the important issue. To confirm our conjecture in more general case, we have added the numerical simulations in Sec.4 for Gaussian mixture models with random mean values and various teacher networks. For all setup we examined, we have found that cRelu STE is similar to the population gradient, while id/Relu STEs show completely different behavior: The result with ten Gaussian mixture input with random mean value and tanh-type teacher network is shown in Fig.1. The behaviors are the same as what we discussed/conjectured theoretically.

---

> > ### Comment · Reviewer_tphS · 2021-12-02
> > **Thank you for the response**
> >
> > Thank you for the response and clarifications. I agree that in Shekhovtsov & Yanush (2020) the noise in the current layer is added just before applying the activation function and is not scaled with the weights. On the other hand, the noise from the previous layers is affecting the inputs of the current layer and is scaled with the weights.
> >
> > My impression after reading the updated manuscript is that it it still not ready for publication. In summary, the authors show that breaking the scale symmetry of the ST estimator is crucial for the considered particularly simple model (one hidden layer, the noise is coming from the inputs only). Their claim that this observation generalises to more complex models remains in my view unverified.

---

### Official Review · Reviewer_W5Me · 2021-11-02

**Correctness:** 2
**Technical Novelty And Significance:** 2
**Empirical Novelty And Significance:** Not applicable
**Recommendation:** 5
**Confidence:** 3

**Main Review:**


I have read the rebuttal. I'd like to thank the authors for the revision, although it does not resolve my concerns.

==========================

## Summary

Training quantized neural network by the biased Straight-Through estimator (STE) is a common practice in the NN quantization literature. However, due to the biased nature of STE, there is no principled guidance in making algorithmic choices for such training methods. This work focuses on the choice of surrogate differentiable functions in STE, and looked at three common choices: identity, ReLU, and clipped ReLU. Through analysis of the scale invariance of gradients in STE, the paper suggests clipped ReLU might pick up local minima degenerated in scales, due to its property of breaking scale symmetry among the three. The authors further analyzed the stationary points of a simple misspecified model with Gaussian inputs to confirm their observations.

## Comments

* The clarity of the paper is very poor. Currently the paper is more like a technical note than a polished conference paper. There are deep stacks of equations everywhere and many of them lack clear explanation. To support my claim, here are a few examples:

Equation (1) is confusing as a convolutional neural network because the hidden layer output is a simple matrix vector product, I figured it out after some thought that you restructured the input by replicating its features in a window to form each row of Z. However, you mentioned this nowhere and it is even not stated what $m$ and $n$ is.

* The main claim is not rigorous, which makes it very difficult to understand the implications of the results and the potential impact of the paper.  Many arguments are hand-wavy and when approximation is used, there is no characterization of the error, to list a few:

"the cRelu STE gradient for |w| << 1 is approximately identical with the Relu STE gradient"

"this relation implies that the Relu gradient is approximately embedded in the cRelu STE gradient"

"may not be zero"

"there is still a chance to be zero"

"using cRelu STE is the most likely to achieve the true minimum in the three STEs"

All these make the conclusion less convincing and the lack of empirical evidence makes it worse.

## Overall suggestions

I vote for rejection mainly due to the questionable analysis (lack of rigor) and the difficulty to interpret the relevance of the result. Also the work does not go beyond one-hidden layer network as it claimed in the abstract.

## Questions and minor points

* Is my understanding correct that the hard $g_j$ has no scale invariance (it only has scale invariance in the loss)?
* I'm always curious, have the quantized neural network community ever looked into other unbiased gradient estimators developed by the Monte-Carlo gradient estimation people? Or is it true that STE empirically outperformed all of them so it is still the default practice?


**Summary Of The Paper:**

Training quantized neural network by the biased Straight-Through estimator (STE) is a common practice in the NN quantization literature. However, due to the biased nature of STE, there is no principled guidance in making algorithmic choices for such training methods. This work focuses on the choice of surrogate differentiable functions in STE, and looked at three common choices: identity, ReLU, and clipped ReLU. Through analysis of the scale invariance of gradients in STE, the paper suggests clipped ReLU might pick up local minima degenerated in scales, due to its property of breaking scale symmetry among the three. The authors further analyzed the stationary points of a simple misspecified model with Gaussian inputs to confirm their observations.

**Summary Of The Review:**

I vote for rejection mainly due to the questionable analysis (lack of rigor) and the difficulty to interpret the relevance of the result. Also the work does not go beyond one-hidden layer network as it claimed in the abstract.

---

> ### Author Response · Authors · 2021-11-23
> **Response to the Reviewer W5Me**
>
> We would like to thank the reviewer for providing the helpful feedback. Our replies to your comment are given below.
>
>        ・The clarity of the paper is very poor. Currently the paper is more like a technical note than a polished conference paper. There are deep stacks of equations everywhere and many of them lack clear explanation.
>
> Thank you very much for your review. We have reduced the equations in Sec.2-3 and added the detailed explanations  in Sec.2.4 in red so as to make our manuscript more clear. Furthermore we also have added numerical experiments in Sec.4.
>
>
> 	・The main claim is not rigorous, which makes it very difficult to understand the implications of the results and the potential impact of the paper. Many arguments are hand-wavy and when approximation is used, there is no characterization of the error, to list a few:
>
> 	"the cRelu STE gradient for |w| << 1 is approximately identical with the Relu STE gradient"
> 	"this relation implies that the Relu gradient is approximately embedded in the cRelu STE gradient"
> 	"may not be zero"
> 	"there is still a chance to be zero"
> 	"using cRelu STE is the most likely to achieve the true minimum in the three STEs"
>
> Thank you very much for the list. We have corrected them in red as much as I could to make the discussion clearer. However, we couldn’t avoid to say “most likely” in the last list, because the behavior always depends on the network architecture and data set, and it is not the theorem but the conjecture as was confirmed analytically in one-Gauss model in Sec.3 and numerically in complicated models in Sec.4.
>
> 	Questions and minor points
> 	Is my understanding correct that the hard  g_j has no scale invariance (it only has scale invariance in the loss)?
>
> Yes, that’s correct. The gradient of the loss g_j does not have the scale invariance, but instead it vanishes in the direction of w due to the invariance of the loss function (the discussion below Eq.7),  while STEs do not have the property (the discussion below Eq.9).
>
> 	I'm always curious, have the quantized neural network community ever looked into other unbiased gradient estimators developed by the Monte-Carlo gradient estimation people? Or is it true that STE empirically 	outperformed all of them so it is still the default practice?
>
> Actually, Bengio et.al (2013) compares the STE with unbiased gradient estimators, and conclude that STE shows the best performance. Also, as explained in Shekhovtsov, Yanush et.al. (2020), the unbiased estimators are known to have serious limitations when applied to deep networks.

---

### Official Review · Reviewer_Tac4 · 2021-11-03

**Correctness:** 3
**Technical Novelty And Significance:** 3
**Empirical Novelty And Significance:** Not applicable
**Recommendation:** 5
**Confidence:** 3

**Main Review:**

Strong points:
* A novel approach on how to analyze various gradient estimators.
* Interesting new theoretical insights into various flavors of the STE gradient approximation.
* Detailed derivations in the appendix.

Weak points:
* No experimental results, even not on MNIST or a toy example to show that the theoretical findings correlate with empirical observations.
* The straight through estimator (STE, identity function) is know to be a biased estimator. In the paper this discussion is not coming back, but I think it would be interesting to discuss whether the other estimators (ReLU, cReLU) are biased and whether one is more biased than the other. Is there a trade-off to be made between being more biased vs having a higher chance to converge to a local minimum?
* The optimization problem defined in 2.1 (eq 1 & 2) is non-convex. For a non-convex optimization problem it is unclear to me based on the manuscript how the gradients in stationary points are important and increase the chance to get to the true global minimum (page 5, below eq 23). Some further explanation on the reasoning could be helpful to understand for the reader and to verify that these claims are true.
* A discussion on whether or not this analysis also holds for 1) weight quantization and 2) multi bit quantization would be interesting. Especially for the latter case it is common practice that the gradient is clipped due to the clipping in quantization (STE is applied to the rounding operation, thus the gradient of the full quantization operation is clipped due to the gradient of the clipping, see Esser et al. 2019).
* The clarity of the paper could improve and there are some misconceptions:
    * The straight through estimator is clearly defined in literature (see Bengio et al. 2013) as back propagation through a function as it would have been the identity function (thus gradient of 1 in the backwards path). However, in the manuscript any approximate gradient is defined as a STE gradient. This misuse of terminology makes it extra difficult and confusing to understand the work, especially since it is of theoretical nature. (“In this paper, we refer to the proxy of the gradient as the STE gradient.”)
    * It is a bit confusing that sigma is defined as step function. But based on the text (binary activations) and later equations it seems that it is a sign function. Would be good to be clear and maybe mathematically describe the exact function.



**Summary Of The Paper:**

The paper does a theoretical analysis of various gradient approximations for the non differentiable sign function. They investigate 3 gradient estimators based on the STE assumption, namely (identity) STE, ReLU STE, and clipped STE (cReLU). They show theoretically that a gradient estimator which breaks the scale symmetry (clipped STE) in the network is more likely to achieve a local minimum than one that keeps the symmetry.

**Summary Of The Review:**

Interesting and novel perspective on various gradient estimators for the sign function used in binary activations. The main drawbacks are that there is no empirical data to support any of their claims, several aspects are unclear based on the manuscript and the paper could improve in its overall clarity. In the current version and my understanding of it the drawbacks outweighs the strong points.

---

> ### Author Response · Authors · 2021-11-23
> **Response to the Reviewer Tac4  (1/2)**
>
> We would like to thank the reviewer for providing the helpful feedback and constructive suggestions. We revised the manuscript according to reviewer’s suggestions. Our replies to weak points you mentioned are given below.
>
> 	・No experimental results, even not on MNIST or a toy example to show that the theoretical findings correlate with empirical observations.
>
> Thank you very much for pointing out the issue. As a first step towards understanding how the theoretical findings are valid for empirical situations, we have numerically tested the Gausssian mixture input with various mean values.  We have found that cRelu STE is similar to the population gradient, while id/Relu STEs shows completely different behavior as shown in Fig.1. The behaviors are the same as what we discussed theoretically.
>
> 	・The straight through estimator (STE, identity function) is know to be a biased estimator. In the paper this discussion is not coming back, but I think it would be interesting to discuss whether the other estimators (ReLU, cReLU) are biased and whether one is more biased than the other. Is there a trade-off to be made between being more biased vs having a higher chance to converge to a local minimum?
>
> Thank you very  much for the useful suggestion. We have mentioned the points in Sec1 (Introduction), Sec2.2 (General discussion), and Sec.4(Experiments) in red as follows:
>
> Sec.1:  … Since the replacement leads to bias, it is not always possible to learn the network successfully. …
>
> Sec.2.2: … This also implies that the cRelu STE is less biased than the other. …
>
> Sec.4: ….We find that population gradient and cRelu STE show similar results, while id/Relu STEs are completely different. This indicates cRelu STE is the less biased estimator than the other estimators. …
>
> 	・The optimization problem defined in 2.1 (eq 1 & 2) is non-convex. For a non-convex optimization problem it is unclear to me based on the manuscript how the gradients in stationary points are important and increase the chance to get to the true global minimum (page 5, below eq 23). Some further explanation on the reasoning could be helpful to understand for the reader and to verify that these claims are true.
>
> Thank you very much for the comments. Our claim is that the cRelu STE is more likely to achieve the “true local” minimum of the loss function than id/Relu STEs. As you mentioned, all the STEs are biased estimators, so that there is no guarantee to show the convergence to the “true” local minimum. They may converge to “fake” minimum which is defined by vanishing point of STE gradient. To avoid confusion for readers, we have changed the sentences near the end of Sec.2.2 to make it clear. We also have added the discussion on biased/unbiased estimators you mentioned above.
>
> 	・A discussion on whether or not this analysis also holds for 1) weight quantization and 2) multi bit quantization would be interesting. Especially for the latter case it is common practice that the gradient is clipped due to the clipping in quantization (STE is applied to the rounding operation, thus the gradient of the full quantization operation is clipped due to the gradient of the clipping, see Esser et al. 2019).
>
> Thank you very much for the interesting suggestion. Actually, we think that the similar discussion is possible for weight binary quantization, because the network is invariant under the scale transformation, while for the multi bit quantization, the network itself breaks the scale symmetry, so our conjecture cannot be applied. (If the network with multi-bit quantization have other symmetries, we can do the same discussion.)

---

> ### Author Response · Authors · 2021-11-23
> **Response to the Reviewer Tac4  (2/2)**
>
> 	・The straight through estimator is clearly defined in literature (see Bengio et al. 2013) as back propagation through a function as it would have been the identity function (thus gradient of 1 in the backwards path). However, in the manuscript any approximate gradient is defined as a STE gradient. This misuse of terminology makes it extra difficult and confusing to understand the work, especially since it is of theoretical nature. (“In this paper, we refer to the proxy of the gradient as the STE gradient.”)
>
> Thank you very much for the comments.  To our knowledge, after the STE is defined as you mentioned in Bengio et al. 2013, in the low-bit network community the term STE is extensively used as the replacement with some differentiable function (see Hubara et al. 2017, Zhou et al. 2016, Esser et al. 2019, Yin et al. 2019) Actually, also in Bengio’s paper, they have studied the replacement with the derivative of the sigmoid. (They refer it  as "a variant".)  To make it clear for readers we have added the explanation about the term “STE” to the introduction as follows:
>
> ….Originally, STE was introduced as the replacement with the derivative of the identity function by Hinton et. al, 2012. Later, the term "STE" has been extensively used as the replacement with various functions. …
>
> 	・It is a bit confusing that sigma is defined as step function. But based on the text (binary activations) and later equations it seems that it is a sign function. Would be good to be clear and maybe mathematically describe the exact function.
>
> Thank you very much for the comment.  To make the definition of σ clear, we have added Eq.(2) in Sec.2.1.

---

### Decision · Program_Chairs · 2022-01-20

**Decision:**

Reject

**Comment:**

I agree with the reviewers that this work is not well-presented, and it seriously lacks rigor and experimental support. The writing of this work also needs significant improvement. The authors made many claims without offering rigorous proofs, and hand-waved their argument throughout without strong empirical support. In the end, the authors' response did not address the reviewers' concerns satisfactorily and no one is excited enough to defend the current draft. Please consider revising your draft according to the reviewers' comments.